# Detection of 85 new active subglacial lakes in Antarctica from a decade of CryoSat-2 data

Sally F. Wilson [1] ✉, Anna E. Hogg [1], Richard Rigby [1], Noel Gourmelen [2], Isabel Nias[3] & Thomas Slater[4]

Subglacial lake activity influences ice sheet flow, grounding line discharge and ice shelf basal melting. Although 146 active subglacial lakes have been detected in Antarctica via ice surface elevation change associated with their activity, only 36 fill-drain cycles have been observed worldwide, and knowledge of these mechanisms is limited. Here, we use a decade of CryoSat-2 radar altimetry to detect 85 active subglacial lakes in Antarctica, documenting 37 and 34 complete draining and filling events respectively. We delineate time-varying boundaries of subglacial lake activity and investigate their variability over time. Our observations increase the number of known active subglacial lakes in Antarctica by 58%, with six of these located within 8 km of the grounding zone. We observe five subglacial lake networks, with concurrent upstream drainage and downstream filling, and 25 clusters of lakes, improving our knowledge of interconnected subglacial hydrological pathways.

Lakes beneath glaciers and ice streams form in subglacial valleys[1], or cavities that evolve as ice slides over bedrock[2,3]. They were first identified beneath the Antarctic Ice Sheet using airborne radio-echo sounding (RES) surveys from the 1960s[1,4]. In Greenland, subglacial lakes can form when seasonal surface melt percolates down from the ice surface to bedrock via moulins, crevasses[5] and hydrofractures[6,7]. Beyond the Antarctic Peninsula[8], there is limited evidence that surface water reaches the bed in this way in Antarctica. Antarctic subglacial water is primarily produced by geothermal heat melting the underside of the ice sheet[9] from pressure-induced melting or through frictional heating of basal ice as it flows downstream[10]. Our understanding of subglacial melt rate magnitudes and how they might vary over time is incredibly limited due to the paucity of measurements in this inaccessible region of the ice base, and as such, subglacial lake activity offers a rare opportunity to measure the timing of change and to quantify episodic water flux at the ice sheet base[11]. Filling and draining events channel water through hydrological pathways between subglacial lakes and have the potential to rapidly influence ice dynamics[12] by modulating basal friction and pressure conditions. If subglacial water reaches the grounding line it can produce buoyant, cold freshwater plumes that increase basal melt rates locally on ice shelves[13],

influence the pattern and strength of ocean circulation[14], and alter glacial landforms and habitats[15]. These effects are especially pronounced in Antarctica where subglacial lakes often occur close to the ice sheet margins[16].

Of the 779 subglacial lakes identified globally, 681 are located in Antarctica, 20% (146) of which have exhibited surface elevation changes suggestive of lake draining and filling cycles[16,17]. An observed cycle refers to a subglacial lake filling event along with its subsequent drainage event. "Active" subglacial lakes, which exhibit these cycles, were first observed in Antarctica in 1997, on the Kamb and Bindschadler Ice Streams, using interferometric synthetic aperture radar (InSAR) data from the Canadian RADARSAT satellite[18]. Clusters of active subglacial lakes have been observed along subglacial hydrological pathways which enable the transfer of water within connected lake networks[12,19–21]. Smith and others[10] used the NASA ICESat laser altimetry satellite mission to identify and delineate the first inventory of 124 Antarctic subglacial lake boundaries based on filling and draining episodes observed between 2003 and 2008. Although laser altimeters have a small footprint size (70 m) and high along-track spatial resolution (-170 m), the spatial coverage is limited by data gaps between repeat tracks which reduces the likelihood that data will be acquired

[1]School of Earth and Environment, University of Leeds, Leeds, UK. [2]School of GeoSciences, University of Edinburgh, Edinburgh, UK. [3]School of Environmental Sciences, University of Liverpool, Liverpool, UK. [4]Centre for Polar Observation and Modelling, Department of Geography and Environmental Science, Northumbria University, Newcastle Upon Tyne, UK. ✉e-mail: eesfw@leeds.ac.uk

over the relatively small subglacial lake features. Further data gaps are caused by the intermittent observational phases that ICESat was operated in and because data collected in the visible part of the electromagnetic spectrum cannot penetrate through clouds. A range of other complimentary airborne and satellite-derived instruments and methods have been used to make measurements of subglacial lake activity. This includes radar interferometry (InSAR)[17,18] which has a high spatial resolution but is limited by the coverage of synthetic aperture radar (SAR) data acquisitions and the methods' reliance on coherence being maintained between SAR image pairs. Stereo-image DEM differencing provides accurate 3D information about ice surface changes. However, this is limited by high data acquisition costs and the impact of noise due to snow and ice reflectance, sky clarity and surface roughness variability[22]. The vertical component of ice surface motion can also be detected using ice velocity measurements from SAR feature tracking[15]. However, these measurements are computationally expensive to produce and have lower vertical accuracy than interferometric techniques. Repeat-pass radar altimetry data has been successfully used to detect active subglacial lakes in isolated cases along the Siple Coast[23–25], at Thwaites Glacier[26–28] on the West Antarctic Ice Sheet (WAIS), and at Slessor[29], Academy[25] and Cook[15] Glaciers on the East Antarctic Ice Sheet (EAIS). This technique provides a consistently processed, long-term record of surface elevation change over subglacial lake features.

Despite efforts to characterise this understudied component of ice sheet mechanics, further work is required to identify the location and extent of subglacial lakes and to characterise the timing and frequency of subglacial lake activity. Currently, the lack of a continent-wide subglacial lake boundary dataset inhibits investigations into their influence on Antarctic Ice Sheet dynamics, in addition to basal hydrology which is currently not represented in continent-scale ice sheet models[30]. Triggers of lake drainage events as well as filling and drainage mechanisms themselves are currently unresolved, and the variability of lake draining and filling cycles over individual lakes is neither well observed nor understood. Over short-time periods drainage events have been associated with ice speed increases of up to 10%[12,27,31] for up to 14 months, with subsequent reductions in ice flow observed. However, the overall influence of subglacial lake activity on ice speed in Antarctica is yet to be determined.

In this study, we use 10 years of swath-processed CryoSat-2 satellite radar altimetry data to detect ice sheet surface elevation change caused by subglacial lake activity. We identify and delineate the boundaries of 85 active Antarctic subglacial lakes and document modes of their variability. We produce an Antarctic-wide observational record featuring multiple cycles of often multi-year episodes of hydrological activity, providing a spatially comprehensive study of subglacial lakes and their evolution over time. We validate our findings by evaluating our method over known subglacial lakes and we compare our results with studies using both data from CyroSat-2 and independent satellites.

## Results

We identify 85 active Antarctic subglacial lakes (Fig. 1), increasing the number of known active subglacial lakes by 58%, from 146 to 231, and the total number of subglacial lakes in Antarctica to 766. We find that 73 of these subglacial lakes are located in East Antarctica, with 12 identified in West Antarctica. Consistent with the majority of active subglacial lakes globally, which are typically found close to ice margins[16], we observe 6 active subglacial lakes located within 8 km of the grounding line and 81% occur under fast flowing ice streams where ice flows > 50 m yr$^{-1}$, (Supplementary Table 1). Our surface elevation change maps show that the active subglacial lake shapes tend to be quasi-circular or elongated (Fig. 2). However, where the subglacial lakes are elongated, we find no correlation between lake orientation and ice flow direction. Although we expect lakes to be aligned with the

direction of subglacial trenches, the impression of subglacial lakes observed at the surface are influenced by attenuation of the height change signal through thick ice, in addition to ice speed, all of which may alter the shape of these height change signals at the ice surface.

Our observations show that 35 active subglacial lakes in this study exhibit monotonic activity during the 10-year CryoSat-2 study period, with 17 subglacial lakes demonstrating just filling and 18 only draining behaviour. Of the 50 subglacial lakes showing both draining and filling, 10 subglacial lakes experience at least one fill-drain cycle over our observational period, two of which, subglacial lakes Whillans_180 and Scott_12, feature two cycles (Fig. 2l, k). During our study period, 37 subglacial lake drainage events and 34 filling events are captured in their entirety (Fig. 2; and Table 1, Supplementary Fig. 2). The median time-period for a complete subglacial lake drainage event in this study is 2.2 years, with the median time for lake recharge being 3.5 years. We define complete events as a subglacial lake fill or drainage episode which is bracketed by the opposite active mode, or a dormant period. We observe the largest drainage event at Cook West_67 subglacial lake, located beneath an ice stream that flows into the Cook Ice Shelf in East Antarctica. This lake drained ~1.3 km$^3$ between early-2012 and early-2016 (Fig. 2f, and Table 1), an order of magnitude larger than the previously reported median Antarctic subglacial lake drainage volume of 0.12 km$^3$ [16]. The Lambert_84 subglacial lake experiences the largest filling episode by volume, filling by ~2.5 km$^3$ from early-2015 to mid-2017, with 8 m of surface uplift observed (Supplementary Fig. 2; and Supplementary Table 1).

Subglacial lake drainage and recharge behaviour is highly variable, both between lakes and for individual lakes themselves. For example, the ice surface over subglacial lake Whillans_180 underwent 5 meters of uplift as the lake filled between 2011 and 2013, followed by a period of surface subsidence from 2013 to 2014 where we observe 5 meters of lowering as the subglacial lake drained to its previous level (Fig. 2l). After filling to approximately the same level for a second time over our study period (1 year duration), this subglacial lake then only partially drains, with 2.8 m of surface subsidence observed before refilling commences (Fig. 2l, and Table 1). In contrast, a number of subglacial lakes also fill and drain to the same level, for example, Totten_52 which was associated with 8 m of both subsidence and subsequent uplift (Fig. 2d). Additional variability is observed in "stepped" filling events (subglacial lakes Totten_52, Byrd_10, and Support Force_3, Fig. 2d, i, and n) and drainage events (subglacial lakes King Baudoin East_3 and Mellor_112, Fig. 2b, c) where we observe quiescent periods during filling or draining, and quiescent periods between draining and filling episodes. Other active subglacial lakes continually cycle through draining and filling. The Moscow University_44 subglacial lake (Fig. 2e) experienced a small drainage event over 6 months in 2017 whilst undergoing longer-term filling. Fill levels vary within individual subglacial lakes, as demonstrated by the time series of surface elevation change over Whillans_180 and Scott_12 subglacial lakes (Fig. 2l, k). Given the length scale of active subglacial lake filling and draining episodes characterised in this study, future studies will require longer-term multi-decadal observational records to improve assessments of Antarctic active subglacial lake behaviour.

Our results indicate that at least 60% of our active subglacial lakes are permanent features rather than transient hydrological signals, as they undergo either a full fill-drain cycle or the opposing mode following a fill or drain event in the same location over the decade long study period. The remaining lakes are likely also permanent features. However, as the study period was not long enough to capture a repeat cycle in these cases, further monitoring will be required to confirm this. Our surface elevation change maps showed that the active region does not always remain constant through time. We delineate time evolving boundaries for each subglacial lake, with a separate boundary for each filling and draining episode (Fig. 2; and Supplementary Fig. 2). The resulting time series of subglacial lake outlines shows that the area

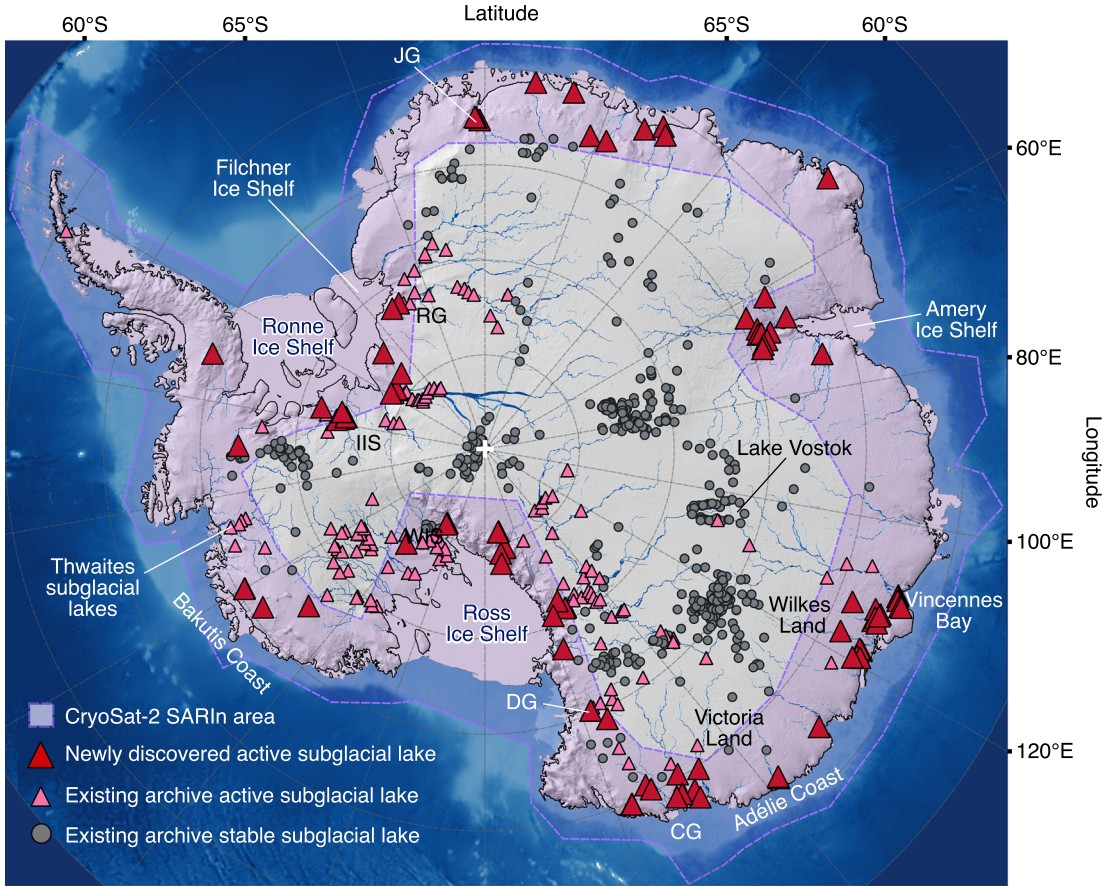

**Fig. 1 | Updated Antarctic subglacial lake inventory.** Antarctic subglacial lake inventory, with 85 lakes identified in this study (red triangles). CryoSat-2 SARIn mode data acquisition region (2015) is illustrated around the continent margins (purple shaded area). Existing inventory subglacial lakes are also shown, divided into those derived from satellite studies (pink triangles) and RES surveys (grey circles)[16,32]. Background image is the CryoSat-2 digital elevation model greyscale virtual hill shade[65] separated from ocean bathymetry (blue background)[66] by the ice sheet grounding line (solid black line)[41,42], overlain with modelled subglacial water routing in blue[14]. Glacier and Ice Stream names are abbreviated: Jutulstraumen Glacier (JG), Cook Glacier (CG), David Glacier (DG), Institute Ice Stream (IIS) and Recovery Glacier (RG).

of subglacial lake activity-induced surface elevation change can evolve by up to 50 %, in the case of two filling events at Scott_12 subglacial lake (Fig. 2k, and Table 1, Supplementary Table 1) where the area of 59.4 km² for the 2014–2016 filling event decreases to 29.8 km² for the later 2018–2020 filling event. The minimum amount of area change we observe between cycles is 9%, for 2 drainage events at Lambert_84 subglacial lake (Supplementary Table 1, Supplementary Fig. 2), indicating large variability on individual lakes. This result highlights the dynamic nature of the subglacial hydrological system in Antarctica and indicates that future studies must assume that this system responds to changes in conditions that vary through time.

In addition to identifying these subglacial lakes, we extended the record of activity over previously identified features in the Antarctic subglacial lake inventories[10,16,32]. We identify 19 lakes in the existing inventory of active Antarctic subglacial lakes[10,26] which are active during our 2010 to 2020 study period and are also located within the CryoSat-2 SARIn mode mask study region (Fig. 1). We assessed the accuracy of the methods used in this study by comparing the timing and magnitude of all documented active subglacial lake episodes during any overlap period with previous studies (Supplementary Table 2). Overall, our results agree with the timing and magnitude of subglacial lake drainage and filling events documented in previous studies. For example, at the Mercer Ice Stream in East Antarctica we observe Subglacial Lake Mercer (SLM) draining rapidly in late 2012 with 7 m of ice surface subsidence, followed by 6 m uplift from a refilling event between early 2015 and early 2018 to almost the same

level as before the previous drainage, before 4 m of rapid drainage over 1 year from early 2018 to early 2019 (Supplementary Fig. 3). These results show good correlation with a previous study on SLM activity with CryoSat-2 and ICESat-2 data[25] in both the timing and magnitude of the dominant filling and drainage events and in shorter-term, small amplitude variability. Beneath the adjacent Whillans Ice Stream, our results show that subglacial Lake Conway fills from late 2010 onwards, followed by a rapid drainage episode in 2019 (Supplementary Fig. 3), a pattern also observed in Siegfried and Fricker's study[25].

The extended observational record shows new modes of activity over two previously documented subglacial lakes: Rutford 1 and Institute E1, both located in East Antarctica, along with a continuation of the previously identified activity at four locations including the Kamb Trunk 1, L78, Cook W1 and Cook W2 subglacial lakes on the Siple Coast (Supplementary Fig. 3, Supplementary Table 2). The latest episode of activity on the Rutford 1 subglacial lake shows a period of draining between late-2010 and mid-2014 where the ice surface lowered by 6 m, followed by 4 m of surface uplift as the lake refilled through to early 2017. Rutford 1 then drained by 3.5 m from early 2017 to late 2018, at which point it refilled through to the end of our time series in 2020 Rough topography over the Rutford Ice Stream introduced large uncertainties into previously assessed CryoSat-2 radar altimetry data[33]. Therefore, elevation change had not been observed over the Rutford_1 subglacial lake since an assessment of the 2003 to 2008 ICESat laser altimetry record[10]. Use of swath processed CryoSat-2 radar altimetry data in this study enabled the Rutford 1 subglacial lake

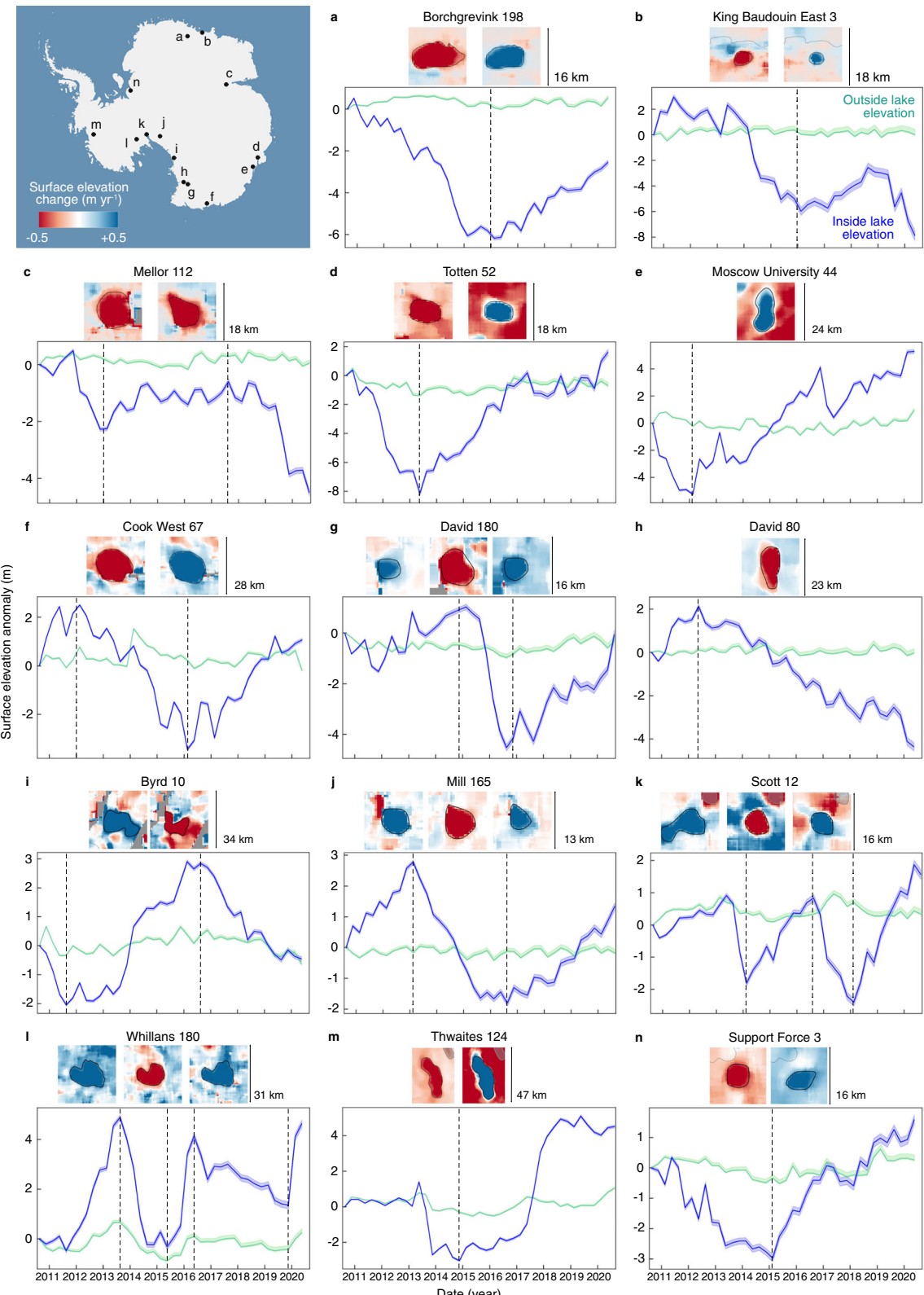

**Fig. 2 | Ice surface elevation change over subglacial lakes.** Three-month mean time series of ice surface elevation change ($dz$) over 13 of the 85 subglacial lakes from this study: **a** Borchgrevink_198, **b** King Baudouin East_3, **c** Mellor_112, **d** Totten_52, **e** Moscow University_44, **f** Cook West_67, **g** David_180, **h** David_80, **i** Byrd_10, **j** Mill_165, **k** Scott_12, **l** Whillans_180, **n** Support Force_3. Subglacial lake elevation change (blue lines) plotted over elevation change from non-subglacial lake regions (light green lines). **m** Results over the previously studied Thwaites 124 subglacial lake for comparison[26,28]. Cumulative error is shown by the shaded area around each time series, and elevation change is relative to the start of each time series in October 2010. Phases of filling and draining for each subglacial lake are indicated by grey dashed lines on the timeseries plots, and the surface elevation change maps for these phases is also shown above. No data areas are indicated by grey pixels in the elevation change maps.

**Table 1 | Record of ice surface elevation change over subglacial lakes in Fig. 2**

| Subglacial lake name | Lat. (°N) | Lon. (°E) | Mean area (km²) | Maximum SEC (m) | Volume change estimate (km³) | Activity log, 2010-2020 |
|---|---|---|---|---|---|---|
| Borchgrevink 198 | -72.57 | 21.48 | 67 | -6.0 ± 0.1 | -0.47 ± 0.008 | **2011–2015 drainage**, 2016–2020 fill |
| King Baudouin East 3 | -70.64 | 28.94 | 16 | -7.0 ± 0.7 | - | **2013–2016** & 2018–2020 **drainage**, 2016–2018 fill |
| Mellor 112 | -74.33 | 66.16 | 76 | -3.0 ± 0.1 | -0.18 ± 0.006 | **2011–2013** & 2018–2020 **drainage**, 2013–2014 fill |
| Totten 52 | -67.58 | 112.62 | 42 | -8.4 ± 0.3 | -0.38 ± 0.014 | **2010–2013** & 2017–2018 **drainage**, 2013–2017 & 2018–2020 fill |
| Moscow University 44 | -67.67 | 118.07 | 115 | +9.1 ± 0.3 | - | 2010–2012 & 2016–2017 drainage, **2012–2016** & 2017–2020 **fill** |
| Cook West 67 | -69.21 | 150.20 | 219 | -5.7 ± 0.1 | -1.31 ± 0.023 | 2010–2011 & 2016–2020 fill, **2012–2016 drainage** |
| David 180 | -74.43 | 155.52 | 44 | -4.0 ± 0.1 | - | 2010–2011 & **2015–2016 drainage**, 2011–2015 & 2016–2020 fill |
| David 80 | -75.17 | 157.93 | 96 | -6.6 ± 0.3 | -0.62 ± 0.028 | 2010–2012 fill, **2012–2020 drainage** |
| Byrd 10 | -80.51 | 157.64 | 155 | +5.0 ± 0.1 | +0.98 ± 0.20 | 2010–2011 & 2016–2020 drainage, **2011–2016 fill** |
| Mill 165 | -85.61 | 170.47 | 26 | -3.6 ± 0.1 | -0.13 ± 0.004 | 2010–2013 & 2016–2020 fill, **2013–2015 drainage** |
| Scott 12 | -85.56 | -152.72 | 46 | +4.2 ± 0.1 | - | 2010–2013 & 2014–2016 fill & **2018–2020 fill**, 2013–2014 & 2016–2018 drainage |
| Whillans 180 | -83.53 | -140.06 | 137 | +2.5 ± 0.05 | +0.45 ± 0.009 | **2011–2013** & 2015–2016 & 2019–2020 **fill**, 2013–2017 & 2016–2019 drainage |
| Thwaites 124 | -76.48 | -106.95 | 346 | +8 ± 0.1 | - | 2013–2015 drainage, **2015–2018 fill** |
| Support Force 3 | -85.56 | -152.72 | 32 | +3.2 ± 0.2 | - | 2010–2015 drainage, **2015–2020 fill** |

Data summary of 13 subglacial lakes from this study, with Thwaites 124 subglacial lake for comparison[26,28]. The subglacial lake naming convention uses the glacier they are located beneath and distance from the grounding line in kilometres[41,42]. In the absence of named glaciers, corresponding ice shelf names are used for nomenclature. Periods of activity highlighted in bold indicate when corresponding maximum surface elevation change (SEC) is recorded, and volume change estimates are computed. Blank volume estimates could not be accurately estimated, due to lake boundaries at least 2 years different from the period of filling or draining. Subglacial lake locations are shown on a map in Fig. 2, along with corresponding SEC maps and time series. Supplementary Table 1 documents all 85 subglacial lakes found in this study and their activity.

activity to be characterised throughout the decade-long study period by retrieving an entire swath of surface height measurements at high spatial resolution, where previously only one aggregated satellite footprint had been observed using the point of closest approach method. Further activity is also observed at the Institute E1 subglacial lake located in Queen Elizabeth Land. Previously recorded filling at Institute E1[33] was briefly halted by a small drainage episode in 2017, after which the lake continued to fill until it drained rapidly from mid-2018 to early 2019, lowering the ice sheet surface by 3.5 m (Supplementary Fig. 3, and Supplementary Table 2). Our results show that long, annual to multi-decadal timescales are required for subglacial lake filling and drainage cycles to be fully captured, and the as-yet unpredictable nature of these events means that updates to this record will be required in the future.

We generated updated lake boundaries for 15 of the previously documented Antarctic subglacial lakes and multiple time-varying boundaries for 12 (Supplementary Fig. 3). Updated outlines are particularly beneficial for subglacial lakes first detected over a decade ago in the period 2003–2009 by ICESat laser altimetry data[10] as the ground track locations and adjacent track spacing provide only partial lake coverage, leading to a more generalised circular boundary delineation. For example, in the David Glacier catchment in East Antarctica, the David 2 subglacial lake boundary was inferred from a single ICESat track (Fig. 4a)[10]. The improved coverage from the swath-processed CryoSat-2 data allows a more elongated sub-glacial lake area to be measured along the north-south axis of the original location, in addition to indicating that the feature is centred 6 km west of the previously delineated boundary (Fig. 4a). An area of surface uplift between two areas of subsidence on David 2 has been identified here via aero-geophysical surveys of the area by Lindzey and others[34], a pattern consistent with our results (Fig. 3a). This study suggests that the observed subsidence is a feature reflecting water flowing in and out of the lake, which is plausible given that both the hydraulic potential gradient and ice flow direction is approximately west-to-east. While it is possible that the lake area that drained in the historical ICESat period is not precisely the same as the area that drained in the more recent 2012 to late-2019 event, it is likely that the higher spatial resolution elevation change data with good spatial coverage did also aid a more accurate delineation of the subglacial lake boundary.

Previous studies showed that the Institute E1 subglacial lake (Fig. 3b) has an area of 460 km² when delineated from five ICESat satellite tracks, with one documented filling episode observed between late 2003 to early 2008[10]. Our results show that a subsequent drainage episode occurred on this lake between 2018 and 2020, with the lake area confined to a smaller, more elongated region which extends downstream (Fig. 3b). We observed an earlier period of filling from 2014 to 2018 which occurred with two main centres of uplift within this elongated area (solid black lines, Fig. 3b). Our surface elevation change results indicate that there is a second centre of activity adjacent to, but distinct from, the main Institute E1 lake area, which we prescribe as a separate lake called Institute 142. Overall, the David 2 and Institute E1 and 142 areas of dynamic subglacial hydrology show the small-scale variability of surface elevation change caused by subglacial lake drainage and filling episodes (Fig. 3), demonstrating the need for high spatial resolution satellite observations to accurately monitor these events. Due to both this and continual improvement to satellite-derived surface elevation datasets, subglacial lake boundaries should be updated over time to improve their accuracy, to document the time-variable regions of change and to avoid under-estimation of surface elevation change magnitudes. Consequently, differences between historical subglacial lake boundaries and those presented in this study (Supplementary Fig. 3) are largely attributable to differences in study time period and satellite data product resolution from which they were produced.

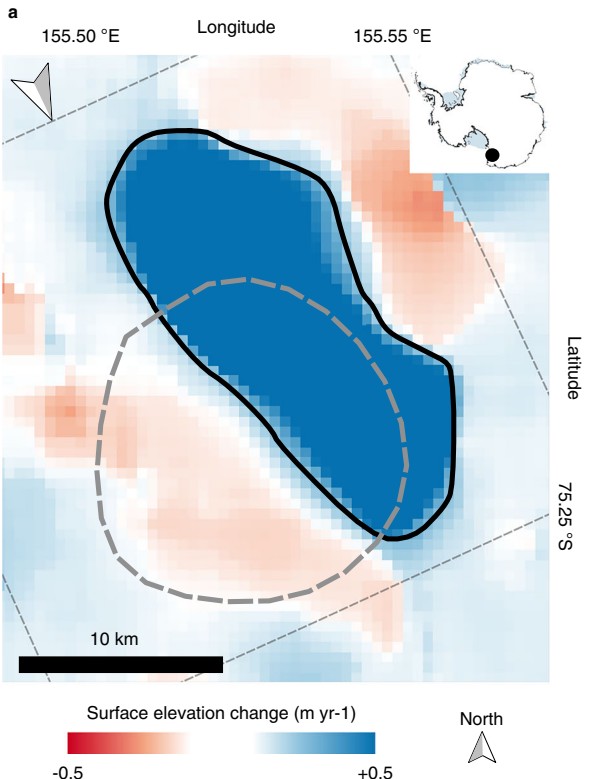
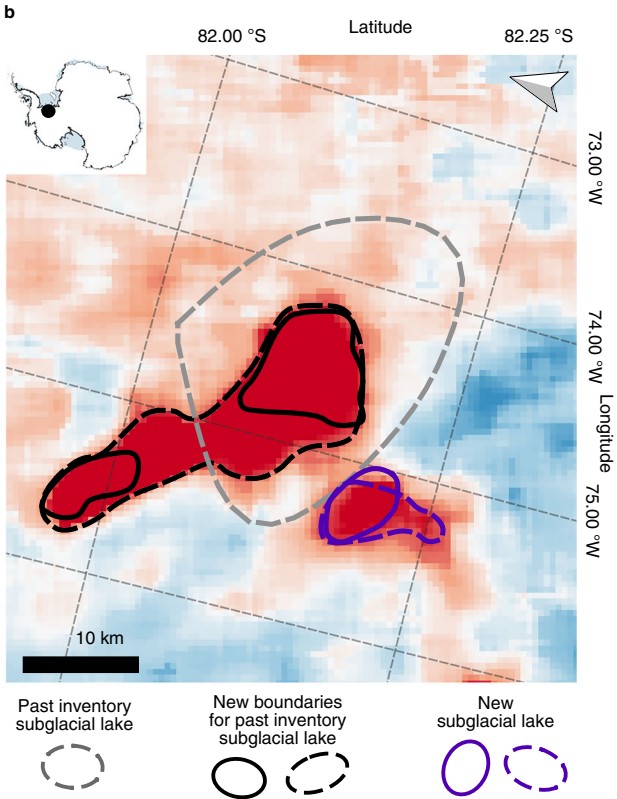

**Fig. 3 | Updated subglacial lake boundaries over David 2 and Institute E1 subglacial lakes. a** Surface elevation change from mid-2010 to mid-2020 over David 2 subglacial lake. The boundary delineated from the 2012 to late-2019 uplift event (black line) is shown alongside the corresponding gridded surface elevation change data in the background. The boundary delineated from the ICESat data[10] is also illustrated (dashed grey line). **b** CryoSat-2 surface elevation change data from early-2017 to mid-2020 over the Institute E1 subglacial lake. The boundary delineated

from this 2017–2020 surface subsidence event (dashed black line) is shown alongside boundaries from a previous period of surface uplift (early-2014 to late-2018) (solid black lines), and the subglacial lake boundary from the ICESat data[10] (dashed grey line). The boundary of a smaller but distinct subglacial lake, named Institute_142 in our study, is located to the South of Institute E1, with boundaries for the 2017–2020 (dashed purple line) and 2014 to 2018 (solid purple line) events also shown. The North direction is indicated in both maps by the grey arrow.

## Discussion

Previous subglacial lake inventories document the existence of active subglacial lakes beneath fast ice streams alongside less active subglacial lakes towards the interior of the continent, often beneath slower flowing ice[1,10,16,33,35,36]. In our study we find that 81 % (69) of our identified active subglacial lakes are located beneath faster flowing ice streams (> 50 m yr$^{-1}$), which is consistent with evidence suggesting that basal meltwater ponding is favourable in areas where ice flows over areas of high basal traction[37], and in areas where the Antarctic Ice Sheet surface is lower, i.e. over larger, fast flowing ice streams[38]. Qualitatively, this statistic may have some bias as our study region is limited to the ice sheet margins where the CryoSat-2 SARIn mode operates (Fig. 1), and this coincides with regions of faster ice flow. Subglacial lake drainage events observed in this study are sustained over relatively long time periods (a median subglacial lake drainage time of 2.2 years and an upper quartile of 4.4 years), which suggests that a large percentage of these subglacial lakes have the potential to affect ice dynamics[39]. Combining our subglacial lakes with those documented in existing inventories provides a more comprehensive picture of subglacial lake distribution in Antarctica (Fig. 1). This is important as subglacial lake dynamics are currently not accounted for in ice mass loss projections, yet subglacial discharge can have significant impacts on ocean melting of ice shelves, ice stream thinning and acceleration[39]. Furthermore, subglacial lake variability detected in this study can inform us on water availability at the ice base, projections of which have thus far been limited due to a scarcity of observations.

The historical subglacial lake inventories suggest that a larger number of active lakes are located in West Antarctica compared to the East Antarctic Ice Sheet[16]. However, our results show that 73 (85 %) of the 85 lakes in this study are located in East Antarctica. This increases the distribution of documented lakes in this region and suggests that active subglacial lakes are more prevalent in East Antarctica than was formerly understood. Our results also suggest that active lakes can exist closer to the grounding line than previously thought. In the historical inventories Lake Engelhardt, beneath the Whillans Ice Plain on the Gould Coast, is the closest active subglacial lake to the grounding line, located ~10 km from the ice sheet margin[40]. Our results show that six subglacial lakes are located within 8 km of the MEaSUREs grounding line[41,42], with the closest two lakes located just 4 km inland of the ice sheet margin. The proximity of active subglacial lakes to the grounding line, particularly those experiencing large or frequent drainage episodes, has implications for grounding line discharge, as subglacial lake drainage events have the potential to increase ice flux through increased speeds[12,31]. The feedback process is two way: retreating grounding lines may also influence the timing or frequency of subglacial drainage events as small changes in surface slope, such as that which occurs during retreat, have the potential to trigger and sustain subglacial lake drainage events[43]. When subglacial lake drainage events occur and water discharges into the ocean, the flux of water can drive high, localised regions of basal melt on ice shelves through turbulent mixing from ocean plumes[13,39,44]. Basal meltwater discharge downstream from subglacial lakes across the grounding line, resulting in these processes, is potentially more likely to occur at grounding line

proximal subglacial lakes, such as that observed 15 km upstream of Muninisen Ice Shelf in Dronning Maud Land[45]. Given that 11 of our subglacial lakes are closer than 15 km to the grounding line, these processing may be influencing ice melt even more than the limited recent observations. Considering the enhanced ice melt and glacier retreat observed in the 21st century, attributed to warming global temperature[46], it will be important to monitor these grounding-line proximal active subglacial lakes in the future as these changes progress.

We observe the five modes of subglacial activity previously reviewed to exist in Antarctica across our dataset: (1) slow filling, rapid drainage, (2) similar filling and drainage rates, (3) rapid filling, slow drainage, (4) quiescent at high stand and (5) quiescent at low stand[16]. We expand this categorisation to include two different modes of activity: (6) monotonic activity and (7) quiescent at mid-stand. We observe 35 subglacial lakes demonstrating monotonic behaviour over the 2010 to 2020 study period. We also observe that subglacial lakes do not always drain to the same level before refilling and vice-versa (Fig. 2b), and there may be a period of stagnation where no filling or draining occurs (Fig. 2c), so this category captures that characteristic. Large variability in both the magnitude and frequency of subglacial lake activity reflects the heterogeneity of their geographical and geometric settings in Antarctica, including factors such as ice thickness and flow speed local to the subglacial lake. Despite evidence that basal hydrology can be highly sensitive to these geophysical parameters[47], an initial assessment suggests that there is no evidence of a correlation with either the four distinct behaviours listed above or mean lake activity, but in the future, further work is required to assess this over all Antarctic subglacial lakes. As complete subglacial lake fill-drain cycles typically last about a decade in Antarctica, it is possible that our 20-year record of ice sheet wide surface observations is currently too short to assess whether subglacial lakes change their modes of behaviour over time. Prior to our study, only 36 lakes with complete fill-drain cycles had been observed globally[16] and our results expand this number to 48. Given the propensity of subglacial lake systems to change dynamically on a time scale of months, future work should extend the observational record of subglacial lake activity in order to improve our knowledge of this short-term dynamic change, so we can improve our knowledge of the physical processes driving subglacial lake activity.

Of 25 lake clusters situated along the same modelled subglacial hydrological pathways[14], indicating hydrological connectivity, our results show that five study areas exhibit upstream drainage events concurrent with downstream filling (Fig. 4). We observe four connected subglacial lakes beneath Lambert Glacier in the Amery Basin (Fig. 4a), three beneath Vanderford Glacier in Vincennes Bay, Wilkes Land (Fig. 4b), two beneath David Glacier in Victoria Land (Fig. 4c), two beneath Stockholm Glacier on the Bakutis Coast (Fig. 4d), and two beneath François Glacier on the Adélie Coast (Fig. 4e). We also identify additional lakes in previously known subglacial lake systems, including the Amery and Aurora subglacial basins, Jutulstraumen, David and Recovery glaciers, Cook and Institute ice shelf tributaries, and Whillans subglacial system. We investigate the connectivity of these networks using water volume change estimates (Supplementary Table 1) to assess whether cascading drainages occur within these subglacial lake systems, where the water from an upstream subglacial lake drains to fill a downstream one. We observe similar volumes of upstream drainage compared to downstream filling on two of these five subglacial lake clusters in East Antarctica. Firstly, a drainage event from 2011 to 2020 on the Vanderford_5 subglacial lake evacuated 0.17 km³ of water, and from 2011 to 2017 (Supplementary Fig. 2) this was observed to fill the Vanderford_4 subglacial lake by 0.12 km³ (Supplementary Fig. 2), located 1 km downstream. Secondly, our results show that a drainage event on the François_46 subglacial lake evacuated 0.05 km³ of water in the 8-years between 2011 and 2019 (Fig. 4b), where the

François_36 subglacial lake coincidently filled by 0.08 km³ of water between 2011 and 2019 (Fig. 4c). Eight years of potential inter-connectivity observed at the François Glacier subglacial lakes, compared to the single year duration (2017-2018) of the coincident drainage and filling of three subglacial lakes in the Thwaites Glacier system[28], indicates decadal variability in connected subglacial systems. Furthermore, the aforementioned cluster at Jutulstraumen Glacier is 81 km downstream of an active subglacial lake cluster observed to fill and drain between 2017–2020[17]. Given their proximity and temporal activity, these lakes may form part of a larger interconnected network and could be associated with cascading drainage. It is possible that our subglacial lake records contain further examples of interconnected hydrological networks, however, as subglacial water may not be fully conserved within a system, these examples can be hard to identify. These results nonetheless provide evidence of subglacial hydrological pathways and reservoir connectivity, which will be useful for understanding the processes that trigger subglacial lake activity.

Beneath David Glacier in Victoria Land, we observe draining of an upstream lake, David_180, coincident with a filling episode ~100 km downstream, at David_80, between 2010 and 2012 (Fig. 4c). Although the upstream lake experiences another 18-month long drainage episode in 2015–2016, the activity of the downstream lake appears to be uncorrelated with this subsequent event. This could indicate that by 2015 the subglacial hydrological drainage system beneath David Glacier has become efficient, to the extent that the ~0.4 km³ subglacial water discharged from David_180 in 2015 does not exceed the hydrological capacity of the subglacial drainage system. Alternatively, it might indicate that the subglacial water routing pathways between these lakes have changed. Thus, David_80 continues to drain at the same rate from 2012 through to the end of our study period in 2020. In Greenland we know that subglacial drainage systems become efficient when large volumes of meltwater are present, such as through seasonal meltwater routing each summer[48,49]. However, limited observations of cascading subglacial drainage events in Antarctica have inhibited our understanding of the drainage system beneath this ice sheet and our knowledge of its ability to reach efficiency, both in terms of spatial extent, water volume and fluxes. Our results show that subglacial lake areas often do significantly change over time, and the events observed on the subglacial lake network under David Glacier indicate that lake connectivity may also change over time. Our results therefore indicate that the location of water routing pathways and subglacial lake network connectivity may be equally dynamic over similar timescales.

More detailed investigations into the dynamics of active subglacial lake networks, for example with a longer time series of elevation change or by harnessing high resolution bed topography data, may provide insights into Antarctic subglacial water routing mechanisms. Channelised subglacial water flow between connected lakes is theorised where water volumes are conserved in cascading subglacial lake drainages[18,50]. Comparatively, a lack of water volume conservation between active subglacial lakes may suggest a broader subglacial network, where water is diverted along several high-pressure subglacial channels rather than ponding in one single location downstream. The active subglacial lake networks observed here therefore provide opportunities for studying these hydrological connections in detail, advancing understanding of subglacial drainage system evolution in Antarctica. Furthermore, case studies of cascading drainage episodes provide unique opportunities for the ice mechanical impacts of subglacial lake fill and drainage events to be assessed in more detail. Combined with records of subglacial water variability, these case studies may help illuminate ice basal frictional changes which may impact ice speed and therefore ice discharge fluxes on both ephemeral and multi-year time scales. Whether we observe subglacial lakes to be situated on or next to modelled hydrological pathways is not so

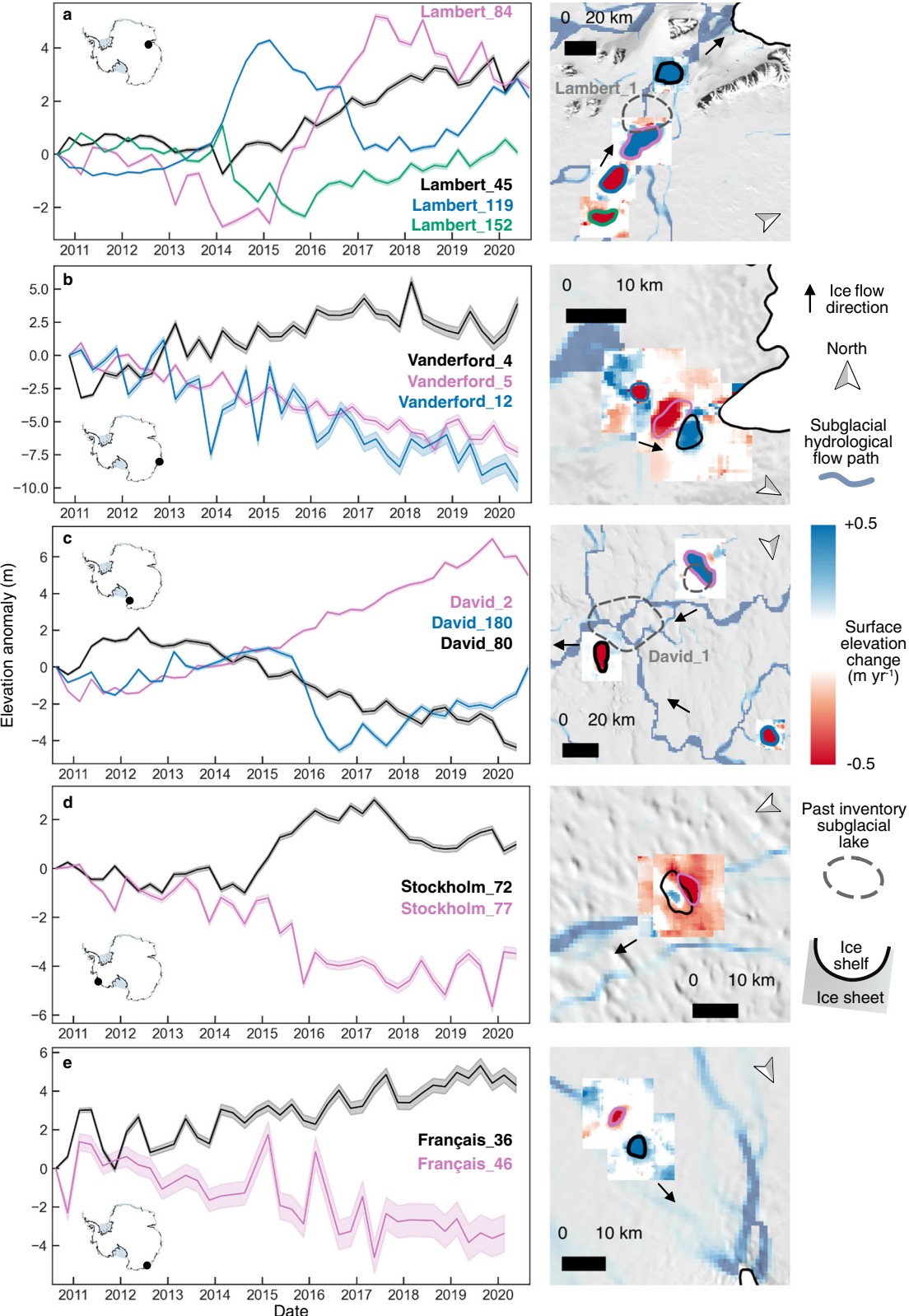

**Fig. 4 | Ice surface elevation change (2011–2020) on 5 subglacial lake networks.** **a** Lambert Glacier, **b** Vanderford Glacier, **c** David Glacier, **d** Stockholm Glacier, **e** François Glacier. Time series of inside-lake elevation anomalies measured for each subglacial lake (left) in the network, with CryoSat-2 surface elevation change maps (right). Derived boundaries (black, pink, blue, green in ascending distance from the grounding zone), and previously identified lakes (grey dashed lines)[11] are illustrated. Subglacial hydrological pathways (blue) modelled from Bedmap2 bed elevation data[14] show a possible hydrological connection between lakes lying along a subglacial stream. Background image is the 125 m MODIS mosaic[67]. Black arrows show approximate ice flow direction.

important, rather simply observing them in a region of lower hydraulic potential where water routing is more likely could provide further evidence for subglacial water fluxes within glaciological catchments.

Active subglacial lake boundaries are dynamic through time[25]. However, with the exception of time-varying outlines published for four subglacial lakes beneath Thwaites Glacier[28], all other subglacial lake boundaries in current literature represent either a snapshot in time, or a long-term average. Static subglacial lake boundaries are unable to capture the full extent of lake dynamics as they are limited to representing long-term, regional height changes. Our results show that subglacial lakes have time-varying area distributions, which is important because it means that their potential impact on ice dynamics may vary spatially as lakes evolve. Ice dynamics dominate mass loss in Antarctica[51], therefore, it is important to quantify any impact from subglacial hydrology. We observe lake area changes of up to 50% and document this in our time-sensitive subglacial lake boundaries, and this provides evidence that subglacial lake dynamics are highly variable over time. Quantifying subglacial lake area change over time will provide useful information for validating ice-hydrology coupled models at high temporal resolution. Documenting the spatially variable pattern of subglacial lake area change and activity may indicate regional sensitivity to bed properties, such as geometry, sediment saturation and temperature, or the volume of subglacial water available. Observing the precise subglacial lake location is also necessary for assessing the potential impact of any associated speed change from ice dynamic processes, or downstream ice shelf melting, which are likely to be highly localised. The observed variability of subglacial lake boundaries over time demonstrates the potential for large-scale subglacial hydrology to evolve dynamically on timescales of single years. Therefore, subglacial lake systems in glacier hydrological catchments must be considered when assessing the influence of subglacial hydrology on ice dynamics, rather than individual subglacial lakes in isolation.

High spatial and temporal resolution satellite measurements of the ice surface, with repeat coverage over subglacial lakes, are necessary for measuring subglacial lake filling and draining patterns. Long term, multi-decadal observational records are required in order to capture complete subglacial lake fill cycles, and even rarer repeat cycles on individual subglacial lakes. This information is essential for improving our understanding of and quantifying how much subglacial lake regimes change over time. These data are also critical for improving our knowledge of the impact of subglacial hydrology on the Antarctic ice sheet system, such as through assessing the sensitivity of feedbacks between subglacial lakes and ice dynamic effects, including both short and long term ephemeral ice speed variability caused by lake drainage and filling events. Basal water discharge at the grounding line from upstream subglacial lake drainage events is thought to impact the localised pattern of ice shelf basal melt[13]. However, further observations are required so we can more accurately quantify the impact of drainage events on ice dynamic processes and ultimately the ice sheet's sea level contribution. The CryoSat-2 SARIn mode mask covers 43% of the Antarctic Ice Sheet (Fig.1), leaving over half of the continent unexplored by this dataset. Hydrological modelling and geostatistics suggest that number of Antarctic subglacial lakes is likely on the order of $10^3$ [52], indicating the existence of many more undiscovered active subglacial lakes that we are not able to image remotely due to the lack of high-resolution satellite observations at higher latitudes. Future satellite missions and studies should therefore extend the record of subglacial lake activity on all documented lakes, in addition to exploring locations where additional subglacial lake networks may exist, by assessing both short and long-term ice sheet surface elevation change signals.

Due to the widespread spatial distribution of active subglacial lakes across the Antarctic Ice Sheet, the bedrock and ice geometries they are located between and interact with vary widely. As the Antarctic Ice Sheet surface is relatively flat, variations in ice thickness, and therefore changes in effective pressure and consequent water motion, are predominantly driven by bed topography. High spatial resolution bed elevation datasets, such as those acquired on geophysical field campaigns and compiled in the Bedmap3 dataset[53], are ideal for assessing the influence of topographic setting and hydraulic potential on water ponding and drainage beneath the Antarctic Ice Sheet. Furthermore, geophysical survey data may contain reflections from subglacial water which could be used to constrain the spatial extent of subglacial lakes from in-situ observations, and to understand the interplay between subglacial topography and subglacial lake lifetimes.

While the likely routing of subglacial hydrological pathways has been modelled[14,54], very little is known about the temporal variability of this water flux. Subglacial lake activity provides one of the few mechanisms for measuring subglacial hydrology variability with any frequency across the whole ice sheet, therefore, we hope these results will be used inform future work in this area, for example, as a validation dataset for high resolution subglacial water routing models. Furthermore, subglacial lake filling and draining activity detected in this study can inform us on water availability at the ice base, projections of which have thus far been limited due to scarcity of observations.

By combining our observations of Antarctic subglacial lakes with the existing inventory, a comprehensive analysis of the subglacial behaviour patterns expanded on in our study can be used to investigate spatial patterns of variability and the physical processes causing them. The significance of short-term subglacial lake variability could also be assessed by comparing sub-annual elevation fluctuations on each subglacial lake to key weather variables, such as localised snowfall variability. A potential connection between sub-annual fluctuations and decadal filling and draining could also be explored. This dataset provides 85 locations over which to assess ephemeral ice speed changes and their potential links with subglacial lakes. Our subglacial lake boundaries and volume estimates also provide case studies for investigating the correspondence between basal and surface dynamics and the effect of ice flow on subglacial signals observed from the surface.

We use CryoSat-2 radar altimetry data to detect 85 active subglacial lakes in Antarctica by measuring elevation change on the ice sheet surface. Our results show that between 2010 and 2020, 37 complete subglacial lake drainage events and 34 complete filling events are observed in our decade-long study period, with 12 fill-drain cycles captured in their entirety. We observe five interconnected subglacial networks with coincident filling and draining events in Wilkes Land, the Amery catchment, Victoria Land, and the Adélie and Bakutis Coasts. Our surface elevation change maps show that the active lake area can change by up to 50% on subglacial lakes that experience multiple periods of activity. We provide the first time-evolving subglacial lake boundary dataset for all 85 lakes, in addition to updating the lake boundary areas of 15 known subglacial lakes in the existing archive. We observe 6 subglacial lakes located within 8 km of the ice sheet grounding zone, which will provide useful information on the impact of subglacial water flux from the ice sheet on other glaciological processes such as ice shelf basal melt rates. Furthermore, subglacial lake locations from this study and their record of activity will provide insight into the impact of these dynamic subglacial systems when assessing the ice mechanical impacts of lake filling and drainage events, quantifying their impact on ice dynamics, and when validating hydrologically coupled ice sheet models. Our results demonstrate the need for continuous, high spatial and temporal resolution measurements of surface elevation change over Antarctica for understanding active hydrology at the ice base, its effect on surrounding ice, and therefore the importance of this process within the Antarctic Ice Sheet system both now and in the future.

## Methods

### Ice sheet elevation change

We use ~15 × 10[9] individual swath-processed[55] elevation measurements over Antarctica from ESA's CryoSat-2 radar altimeter satellite mission to investigate ice surface height changes for the period October 2010 to July 2020 (Supplementary Fig. 1). This enables us to identify areas of spatially consistent height change indicative of subglacial lake activity. With a latitudinal limit of 88 ° and a monthly sub-cycle, operating in Synthetic Aperture Interferometric (SARIn) mode over the ice sheet margins[56], the CryoSat-2 radar altimeter instrument acquires surface height measurements at ~400 m horizontal spatial resolution over Antarctica. Swath-processing utilises phase information to unwrap the data across the track, providing a swath of elevation data rather than a single measurement at the point of closest approach (POCA)[55]. This yields a substantial increase in the spatial density of the altimetry datapoints, which in turn enables the subsequent surface elevation change processing steps to be run on grids at a 10-times finer spatial resolution than possible using traditional approaches. This makes swath processed CryoSat-2 data ideal for detecting relatively small subglacial lake signals around the Antarctic Ice Sheet margins.

We separate elevation fluctuations evolving over time from those resulting from local topography and temporal variations in radar backscatter within 500 × 500 m grid cells using a model fit method[57–59]. Firstly, we remove unreliable swath elevation measurements where the coherence value is < 0.8, the scaled power is < 10,000 and the difference with respect to a reference DEM[60] is > 25 m. We model the elevation $z$ as a function of local surface terrain $(x, y)$, time $(t)$, satellite heading $(h)$ and the mean height within each grid cell $(\bar{z})$:

$$z(x, y, t, h) = \bar{z} + a_0 x + a_1 y + a_2 x^2 + a_3 y^2 + a_4 xy + a_5 h + a_6 t \quad (1)$$

The satellite heading $(h)$ is a binary term set to 0 or 1, based on whether the satellite track is ascending or descending, respectively. We solve Eq. 1 within each grid cell using an iterative least-squares fit to minimise the impact of outliers and discard poorly constrained solutions based upon the following set of statistical thresholds: a minimum of 10 time points, a time series length of at least 2 years, a maximum root mean squared difference of elevation residuals from the model of 12 m, a maximum elevation rate magnitude of 10 m yr$^{-1}$, and a maximum surface slope of 5 °[58]. We compute the maximum surface slope from static BedMap2 ice surface elevation data[60].

We account for spurious variations in range associated with changes in radar echo shape by applying an empirical correction based upon correlated changes in elevation and backscattered power, consistent with existing CryoSat-2 SARIn altimetry studies[58,59]. To do this, we compute the correlation gradient in elevation as a function of power, $dz/dp$, using a linear fit in each grid cell. The change in backscattered power $dp$ is multiplied by this correlation gradient and then subtracted from the model elevation anomalies to produce power-corrected anomalies[59]:

$$dz_{corrected} = dz - \left( dp \frac{dz}{dp} \right) \quad (2)$$

The elevation anomalies are averaged over 90-day epochs to produce mean elevation change time series within subglacial lake boundaries and their surrounding area. We estimate the uncertainty in our regional elevation time series at each epoch by computing the average of the standard error $(\sigma/\sqrt{n})$ of the model elevation anomalies within all contributing grid cells[57,58]. We assume this component is temporally uncorrelated, and for each epoch we then accumulate all preceding uncertainties in quadrature[57].

### Subglacial lake boundaries and elevation change time series

We use our 10-year elevation change map (Oct 2010–July 2020) to identify potential subglacial lake points of interest (POI). To maximise the likelihood of ice surface height change reflecting real subglacial lake activity we pinpoint areas of spatially consistent surface height change ≥5 km that are distinct from measurement noise. Time series are first extracted from a 5 km box centred around each active subglacial lake POI, and a surrounding box 20 km in either direction which excludes the subglacial lake area, which we class as the non-subglacial lake region (Supplementary Fig. 1a). Each mean elevation data point is normalised relative to the first timestep's elevation to yield a final elevation change time series. We remove any large scale regional surface elevation change signals not linked to subglacial lake activity, such as from ice dynamic thinning[57], by calculating the 10-year linear elevation trend calculated from the 20 km non-subglacial lake box region and subtracting this from the subglacial lake region. These time series are then compared to determine locations featuring distinct elevation change above the background non-subglacial lake elevation change (Supplementary Fig. 1b), which we interpret to be periods of significant lake activity.

We note that processes such as the viscous deformation of ice or the impact of ice flow may alter the surface impression of behaviour at the ice bed[61]. The use of "subglacial lake boundaries" in this manuscript refers to the surface impression of elevation change observed due to subglacial lake filling and draining, rather than a direct delineation of the physical subglacial water edge. In order to measure the maximum extent of ice elevation change over each subglacial lake, we produce a new surface elevation change map for the discrete period of surface uplift or subsidence as determined from the elevation change timeseries. Surface elevation change maps produced from only part of a filling or drainage event cycle will result in a partial elevation change signal in the gridded maps. In the event that the map is produced over the full period of the cycle, there will be zero elevation change visible in the gridded data making it impossible to accurately delineate the subglacial lake boundary. We use the maximum elevation change maps to manually delineate boundaries for each subglacial lake, producing a separate lake boundary for every filling and draining event (Supplementary Fig. 1c). We draw the lake contour at the ± 0.3 m yr$^{-1}$ elevation change threshold, based on noise sensitivity testing, and we adjust this depending on the signal magnitude, altimetry data coverage, and elevation patterns compared to the surrounding ice. Time series of ice surface elevation change are then re-extracted from within each subglacial lake boundary, including pixels whose centres fall within the boundary. We adjust the non-subglacial lake box size to equal the maximum subglacial lake diameter plus 5 km, before also re-extracting time series of surface elevation change for each non-subglacial lake region. All time series are once again detrended using a newly calculated 10-year long linear trend and the subglacial lake active periods are updated based on this second processing iteration with the more accurate lake boundary information. We perform a final processing iteration by smoothing each subglacial lake elevation change map using a median filter with a kernel size 10 % of the lake polygon area or 2 km$^2$, whichever is smaller. We compare the raw and smoothed data to ensure they agree, and we update the lake boundaries with the smoothed data in regions where noise prevented a continuous outline being drawn from the raw data exclusively. Following this final iteration, we re-extract the elevation change time series from each active subglacial lake. We assess uncertainty in our subglacial lake boundary delineation technique by drawing boundaries for three subglacial lakes, five times each[62], to represent variability with subglacial lake size: King Baudouin West_51 (19.74 km$^2$), Mill_165 (35.8 km$^2$) and Borchgrevink_198 (78.49 km$^2$) (quartile lakes by area). The pooled standard deviation of the subglacial lake area measurements is

0.453 km$^2$, which reflects the 500 × 500 m spatial resolution of the swath-processed altimetry dataset. Therefore, we assume this standard deviation to be the uncertainty on our subglacial lake area measurements.

## Volume change estimation

To estimate the magnitude of water movement beneath the ice sheet from this subglacial lake activity, we calculated the volume of surface ice displaced for each filling and drainage event. We note that equating these two volumes varies in accuracy as a function of ice thickness, subglacial lake drainage rate and basal drag coefficient[61]. We therefore perform a simple volume calculation (Eq. 3) and suggest coupled ice-hydrology models for the subglacial lake networks presented in this study for future work, in order to assess variability in the surface-bed relationship. We do not provide volume change estimates for 79 cases (53%) where the subglacial lake boundary was not delineated for a discrete period of activity. This usually corresponds to more rapid (less than 2 years) filling or drainage events where poorer altimetry data coverage leads to higher noise levels in the surface elevation change maps, making it hard to delineate the lake boundary. Volume change, $dV$, is estimated using lake area, $A$, and mean surface elevation change, $dz$, from each time period of lake activity, $t$, as follows:

$$dV_t = dz_t \times A_t \qquad (3)$$

## Data availability

The subglacial lake shapefiles and surface elevation change time series data generated in this study have been deposited in the Zenodo database under accession code (https://doi.org/10.5281/zenodo.16330564)[63]. The metadata on subglacial lakes and their activity generated in this study are provided in the Supplementary Information/Source Data file. The CryoSat-2 satellite altimetry data used in this study are available from the European Space Agency in the CS2EO database (https://cs2eo.org/). Antarctica base map data used for map insets are from the SCAR Antarctic Digital Database (ADD) Version 7.0., with maps produced in Quantarctica[64].

## Code availability

The above-referenced repository contains code to reproduce elevation change time series over each subglacial lake presented in our study. The altimetry processing chain is available from Slater and others (2021)[58].

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

## Acknowledgements

This work was led by the School of Earth and Environment at the University of Leeds. Data processing was undertaken on ARC3, part of the high-performance computing facilities at the University of Leeds. The authors gratefully acknowledge the European Space Agency for the acquisition and availability of CryoSat-2 data. The authors were supported by the Natural Environment Research Council (NERC) Panorama Doctoral Training Partnership (DTP), under grant NE/S007458/1 (S.F.W.), the European Space Agency 4DAntarctica project 4000128611/19/I-DT (N.G. and A.H.), the NERC via the DeCAdeS project NE/T012757/1 (A.E.H.) and the UK EO Climate Information Service NE/X019071/1 (A.E.H.).

## Author contributions

S.F.W. and A.E.H. designed the work. S.F.W. wrote the manuscript, performed the analysis with guidance from A.E.H. and I.N., and produced the subglacial lake boundary dataset. N.G. produced the Swath mode CryoSat-2 elevation data, R.R. processed the elevation change measurements based on code by T.S., S.F.W., A.E.H., T.S., I.N., N.G., and R.R. contributed to scientific discussion and revisions of the manuscript.

## Competing interests

The authors declare no competing interests.
