## [Transparent Peer Review file · Nature Communications]

Detection of 85 new active subglacial lakes in Antarctica from a decade of CryoSat-2 data

Corresponding Author: Ms Sally Wilson

Version 0:

Reviewer comments:

Reviewer #1

(Remarks to the Author)

Review of manuscript "Detection and variability of 85 new active Antarctic subglacial lakes with CryoSat-2 radar altimetry, 2010-2020" by Sally F. Wilson, Anna E. Hogg, Richard Rigby, Noel Gourmelen, Thomas Slater, and Isabel Nias. Submitted to Nature Communications.

Overall, I found the manuscript to be well-written and well-structured, and I enjoyed reading it. The results are important and advances current knowledge on subglacial hydrology in Antarctica. There are some elements that are less well explained, and some methods and arguments that should be either revised or justified better. These are described below.

Key results:

The study uses CryoSat-2 swath-processed radar altimetry data to monitor surface elevation changes over Antarctica between 2010 and 2020, identifying 85 new active subglacial lakes. A key contribution of the study is the creation of a time-evolving dataset of subglacial lake boundaries, capturing large variation in active lake areas. The study findings provide important insights into the subglacial water flux beneath the Antarctica Ice Sheet.

Validity:

While the detection of new lakes is very timely and interesting and the identification of activity patterns is promising, I have a few concerns regarding the interpretation of some of the results.

Specifically, the delineation of subglacial lake perimeter outlines from surface change signals is potentially overinterpreted. The assumption that surface expressions reflect the true subglacial lake boundaries is not adequately justified in the current version of the manuscript. Previous work (e.g., Stubblefield et al., 2021) has shown that surface depressions may not reliably indicate subglacial lake extents due to the influence of viscous ice flow, and this should be addressed more specifically. Maybe it is simply a question of the terminology, so that you do not claim to delineate subglacial lake perimeters but rather subglacial lake expressions or signals?

Additionally, the volume estimates are derived from a simple equation (Eq. 3) that lacks detailed justification, and it is unclear if this method meets current standards in the field (Stubblefield et al., 2021).

Significance

The study presents a significant expansion of the known inventory of active subglacial lakes in Antarctica and it offers a rare, decade-long dataset of lake activity. This contributes to our understanding of Antarctic subglacial hydrology, which is a topic that is important and timely.

However, the significance of the volume estimates is less clear, especially given uncertainties in methodology. The most impactful findings relate to the detection, variability, and spatial distribution of the lakes rather than their calculated volumes. Therefore, I would recommend that the authors reconsider if the lake volume estimation actually adds anything crucial – and if not, that they take that part of the study.

Impact: I would like to see the authors elaborate on the impact of their study. How exactly will our understanding be improved by the detection of these new active subglacial lakes? I see one obvious one, being the fact that basal melt is currently neglected in mass balance studies of the AIS. And as you state in L 30-31 the SGL variability can inform us on the water availability at the base.

Data and Methodology

The use of CryoSat-2 radar altimetry to track surface elevation changes is appropriate and well established. The spatial and temporal resolution is very suitable for the research objectives.

However, I find that parts of the methodology needs improved explanation.

The approach to assessing uncertainty in subglacial lake delineations (manual drawing of a few lake perimeters is simplistic and I believe it does not fully address an important source of uncertainty: the mismatch between surface signal and actual lake geometry. This needs to be made clearer throughout the manuscript.

Further with respect to the delineation; outlines are provided less frequent than the elevation change results. How come? I would think that the outlines would change continuously and that more regular outlines (and hence areas) are needed for the volume calculation. Here, I am especially thinking of the outline change caused by ice deformation.

When removing outliers (L. 88) a threshold is applied on how far from a reference DEM the altimetry data can be to be included. The threshold is 25 m. Does this mean that you will not be able to detect surface depressions deeper than 25m? Does this not potentially filter out some subglacial lake variability signals?

Can you be more specific about what the standard error of the height change is? How is that derived? (L. 101).

Analytical Approach

The analytical approach is generally sound in terms of identifying elevation change events and characterising their timing and extent. However, there are a few things that could be improved:

Please elaborate on the correlation described in Eq. 2. It is not clear to me why you would expect such a relationship in swath-processed altimetry data. The Davis et al., (2004) reference is for a study using conventional RA from ERS-2, using a retracking algorithm that can be sensitive to correlations between the waveform and estimated height.

As mentioned above, the volume estimation via Eq. 3 requires additional justification and comparison with more advanced techniques. Reference to recent work such as Stubblefield et al. (2021) is warranted and should be included to contextualize the limitations of the current approach.

Further suggested (mostly technical) improvements:

- The current title, "Detection and variability of 85 new active Antarctic subglacial lakes with CryoSat-2 radar altimetry, 2010–2020", appears somewhat awkward in phrasing to me. Particularly the construction "variability ... with" may suggest that variability is being performed using the instrument, which is conceptually unclear. I recommend rephrasing for clarity and grammatical correctness. For example: "Detection of variability of 85 new active Antarctic subglacial lakes from CryoSat-2 radar altimetry (2010–2020)".
- L 14: what is meant by worldwide here? You are only discussing subglacial lakes in Antarctica.
- L 16 do you mean complete cycle or complete drainage?
- L 17 at this stage it is not clear what a cluster or a network refers to
- L 22 I would change valleys to depressions.
- L 37: provide exact number instead of percentages.
- Equation 1: I believe that you have not explained all the terms in the Eq.
- L 174-175 I would suggest to not provide the average time periods as I do not see how these numbers provide any useful insight, since these lakes are located in different regions and have different driving processes.
- Table 1: Please include uncertainties
- Fig 1. The scale number for Mill 165 is partly hidden behind the figure.

References:

Davis, C. H. & Ferguson, A. C. Elevation change of the Antarctic ice sheet, 1995-2000, from ERS-2 satellite radar 573 altimetry. *IEEE Trans. Geosci. Remote Sensing* 42, 2437–2445 (2004).
Stubblefield, A. G., Creyts, T. T., Kingslake, J., Siegfried, M. R., & Spiegelman, M. (2021). Surface expression and apparent timing of subglacial lake oscillations controlled by viscous ice flow. *Geophysical Research Letters*, 48(17), e2021GL094658.

(Remarks to the Author)

For full review comments, please see attached pdf.

[Editorial Note: The attachment has been appended to the end of this file]

General comments:

This manuscript uses CryoSat-2 radar altimetry swath-processed surface elevation data to develop a new dataset of active subglacial lake boundaries and time series of their filling and draining over a decade (2010-2020). The authors document complete filling and draining cycles and identify a number of interconnected subglacial lake networks. They found a total of 85 active subglacial lakes and five interconnected lake networks and conclude that individual lake areas can change by up to 280% over multiple periods of filling and draining.

Hydrologically active lakes are an important, yet poorly constrained component of the ice sheet subglacial hydrological system. Active lakes can transport excess meltwater episodically towards the grounding line, where it is discharged into sub-ice shelf cavities and can enhance ice shelf basal melting. Subglacial hydrology beneath the Antarctic Ice Sheet remains poorly understood and challenging to represent in ice-sheet models. Observing active lakes using repeat satellite data is therefore crucial to understanding the impact of subglacial hydrology on ice dynamics and the ice sheet-ocean system. Few studies have produced ice sheet-scale inventories of active subglacial lakes, and studies using different satellite sensors each have their own limitations, e.g. limited coverage.

It is my view that this study is of broad interest to the cryospheric community as it builds upon previous work focused on subglacial lake detection beneath the Antarctic Ice Sheet, especially in the context of subglacial hydrology and ice-sheet dynamics, by extending the record of active subglacial lake activity and their time-evolving filling and draining behaviour. I think the ice sheet modelling community would be very interested to use this dataset as validation at an ice-sheet scale in future. Overall, this is a well-written manuscript which could be clearer some places, detailed by my specific comments. Once the authors address these, I can therefore recommend that this manuscript is suitable for publication in Nature Communications.

I have a few general points, which I outline below:

- In the abstract and several other places in the manuscript, 5 new subglacial lake networks and 25 clusters are mentioned. However, it is unclear to me what these 25 clusters refer to – are these lakes that intersect with modelled subglacial water flux (LeBrocq et al., 2013)? From Figure 1, it is unclear which of the 85 new lakes constitute these 25 clusters, and it is difficult to see in any detail the relationship between the authors' new reported active lakes and modelled subglacial water flux. This latter point is not really elaborated on in the manuscript.
- The manuscript does a great job of validating lake outlines using previously reported lakes derived from other sensors. I would like to suggest that the results of our recent study (Arthur et al., 2025) would be a valuable addition to this comparison, as it provides one of a few records of active subglacial lakes near the grounding line with several lakes coincident or located near those identified in this study. Specifically, locations such as Lazarevisen (coincident with our Lake L1), Vigridisen (46 km upstream of our lake V1), Muninisen (170 km upstream of our lakes M1 and M2) and Roi Baudouin (78 km downstream of our Lake R2) could provide useful points of comparison. I have included two of these examples below. In particular, relevant discussion could be added to the intercomparison of subglacial lake inventories (Section 4.1) and in Supplementary Figure 3.
- In the Discussion (Section 4.3), I suggest mentioning the lake cluster upstream of Jutulstraumen (Jutulstraumen_69, _71, _74), which you detect filling and draining between 2013–2015 and 2017–2020. This cluster is located 81 km upstream of the active lake cluster reported by Neckel et al. (2021) using Sentinel-1 DInSAR, validated with ICESat-2, and observed to fill and drain between 2017–2020 (see right-hand map). Given their proximity and temporal activity, these lakes may form part of a larger interconnected network and could be associated with cascading drainage.
- One of the interesting findings from this new inventory is the detection of active lakes near the grounding line in certain locations (e.g. King Baudouin East_3, less than 4 km from the GL). I think this deserves further emphasis (for example, an additional figure highlighting some of these new examples) due to its important implications for subglacial water routing and the impact on local ice dynamics at the transition between grounded and floating ice in these regions.
- As a general point, some of the discussion about future work suggestions could be more suited to Section 4.5 (Future Outlook), for example the last few sentences of Section 4.1 and 4.2.
- It is good to see details of the subglacial lakes, drainage/filling periods and associated area and volume changes provided in the Supplementary Data. I would encourage the authors to consider making their code available to reproduce the results presented in this manuscript, in line with Nature Communication guidelines and to promote reproducibility.

Reviewer #3

(Remarks to the Author)

The authors present a new inventory of Antarctic subglacial lakes derived from CryoSat-2 altimetry data extending current

inventories by 85 (+ 58%) newly detected lakes. These newly detected lakes were mapped in higher spatial resolution than previously detected lakes by laser altimeters and the presented time series on lake variability allow detailed analysis on lake dynamics. For every newly detected lake, the authors present time series on lake drainage and fill events between the period 2010-2020.

Even though, I am not an altimetry expert the methodology employed appears to be both robust and comprehensively delineated. Especially making use of the POCA processing approach allows a much finer spatial resolution for lake mapping. Therefore, this study presents a much more detailed and accurate dataset compared to the existing inventory as can be seen in Figure 3. It is positive to note that the shapefiles of the newly detected lakes have been published open access. The text is clear and accessible, with results presented in a well-structured manner.

The significance of the work is given in any case due to the importance of Antarctica's subglacial hydrology for ice loss (Fricker et al. 2025). Nevertheless, for the scope of Nature Communications I would have expected more insights regarding the process understanding of subglacial lake activity and distribution. The authors define lake variability and behavior into seven categories (L 351 ff.) and provide insights into connected subglacial networks (L 367ff.). But the data produced within this study would facilitate a greater level of insight with minimal effort. The following ideas could be pursued:

- The supplementary includes all time series of the 85 analysed lakes. There seems to be a strong seasonality in the drainage and filling events. Did you have a look into the seasonality (e.g. detrending and performing a seasonality analysis on time series with sufficient measurement points). Why is there strong variability in some years but not in other?
- Subglacial lakes seem to be clustered in some specific areas (Figure 1). Can you explain why?
- The lakes are categorized into seven categories. How is their spatial distribution? Have you tried to create a map based on these categories and where you able to identify any spatial patterns?

Specific comments:

Figure1: Add the CryoSat data acquisition limit at the South Pole to the map to show where potential active lakes might have been detected.

Table 1 and L 139 ff. Can you provide any uncertainties for the volume change estimate? In L 157 you mention that the height change signal at the surface is attenuated by thick ice and hence the change in lake level is very likely different from the surface elevation change and might introduce large errors.

Table 1: What is the reason for your choice of these 13 lakes in particular?

L174: Also consider providing the median and quantiles for drainage events. Just the mean average could be miss leading in case of strongly varying time spans.

L 328: You mention that subglacial lakes can exist closer to the grounding line (GL) than previously thought. A recent study by Arthur et al. 2025 also mentions subglacial lakes close to the grounding line. Consider discussing these findings with yours.

L332: You used the MEaSURES grounding line. Did you compare the distance between the GL and your detected lakes with a newer grounding line product?

Figure 4: Here you present the drainage of Francais_46 and subsequent filling of Francais_36. Did you do any statistical analysis on the sub-annual fluctuations and the connection between drainage and filling? For example, in 2015 F_46 re-fills and drains half a year later. Do you see in return a filling signal in F_36 with a temporal lag?

Supplementary: There seem to be some lakes that experienced a surface lowering but lake area expansion (e.g. Jutulstraument_71, Fisher_168). Is this a common pattern and what is the explanation behind?

References

Arthur, J.F., Shackleton, C., Moholdt, G., Matsuoka, K., van Oostveen, J., 2025. Evidence of active subglacial lakes under a slowly moving coastal region of the Antarctic Ice Sheet. *The Cryosphere* 19, 375–392. <https://doi.org/10.5194/tc-19-375-2025>

Fricker, H.A., Galton-Fenzi, B.K., Walker, C.C., Freer, B.I.D., Padman, L., DeConto, R., 2025. Antarctica in 2025: Drivers of deep uncertainty in projected ice loss. *Science* 387, 601–609. <https://doi.org/10.1126/science.adt9619>

Version 1:

Reviewer comments:

Reviewer #1

(Remarks to the Author)

Dear authors,

Thank you very much for your response and the revisions you have made to the manuscript.

I have still a few outstanding issues that I would you to address. I refer to the numbering used in your rebuttal.

1.2: Thanks for the revisions. I acknowledge your explanation which is fair – but I would suggest that you rephrase the statement in the abstract to make it clear there as well.

#1.8: I think that the threshold applied by Slater et al., (2021) is very appropriate given the aim of that study. But in this study the aim is different, and I do not understand why it is needed to have such a threshold. In McMillan et al., (2013) it is stated about the Cook_E2 lake that "Between November 2006 and March 2008, the ice surface fell by ~ 45 m" so isn't that an example of a signal that you would miss?

1.10: I have read now the three mentioned papers, and I apologize if I have misunderstood something, but I believe that neither Slater et al., (2021) or McMillan et al. (2014) used swath-processed data as you do here. Further, if I understood correctly, the power is indeed used in Gourmelen et al., (2018) but not for surface elevation change correction but for weighing individual elevation points from the swath, which is not what I understand that you are doing here. Please elaborate to justify this approach.

Reviewer #2

(Remarks to the Author)

I have reviewed the revised manuscript and feel that the authors have addressed the comments from myself and the two other reviewers effectively. I'm pleased with the improved clarifications of terminology, and extended descriptions where requested.

I only have a couple of minor comments:

- I think it could be helpful for the reader to have a key in Figure 3 (as in Figure 4).
- In the Abstract, I appreciate the authors have tried to clarify their definition of what constitutes individual lake clusters, but this could be interpreted as 25 lake clusters located within (the same) small geographical region. I would therefore remove this from the abstract and instead add this to in the Discussion ('Connected Subglacial Lake Networks').
- I appreciate the authors' explanation of their selection of the ± 0.3 m yr⁻¹ elevation change threshold in their response to my comments, and I think it would be useful to include this in the main manuscript (L129).
- It may be worth adding a caveat that, while you observe that 25 lake clusters appear aligned along the same modelled subglacial hydrological pathways (LeBrocq et al., 2013), and this is used to infer hydrological connectivity, the underlying water routing methodology is based on the Bedmap2 grid. This grid can be significantly smoothed between radar profiles, meaning that flow routing is primarily influenced by topographic features near the closest radar tracks. As a result, for lakes located farther from these profiles, the inferred locations of subglacial streams may be overly confident or less reliable. For more info see Shackleton et al. (2023, <https://doi.org/10.1029/2023JF007269>).

Reviewer #3

(Remarks to the Author)

I appreciate the authors for their thorough responses and clarifications to all reviewer comments. It's creditable to see the manuscript enhanced in various aspects, particularly with the addition of Figure 4, which significantly contributes to the comprehension of lake clusters and their interconnections. I recognize that some of my questions related to lake dynamics necessitate further analysis that fall outside the scope of this paper. Therefore, I look forward to the authors' future publications that will likely explore the dynamics of subglacial lakes, including the potential seasonality in filling and drainage events.

Response to referees: Detection and variability of 85 new active Antarctic subglacial lakes with CryoSat-2 radar altimetry, 2010–2020

Ref: NCOMMS-25-13112

We thank each reviewer for their positive and constructive comments. Please see a response to each comment addressed in the table below.

Reviewer 1:

ID	Comment	Line	Response
1.1	Overall, I found the manuscript to be well-written and well-structured, and I enjoyed reading it. The results are important and advances current knowledge on subglacial hydrology in Antarctica. There are some elements that are less well explained, and some methods and arguments that should be either revised or justified better.		We thank reviewer 1 for their time, positive comments and helpful feedback.
1.2	The delineation of subglacial lake perimeter outlines from surface change signals is potentially overinterpreted. The assumption that surface expressions reflect the true subglacial lake boundaries is not adequately justified in the current version of the manuscript. Previous work (e.g., Stubblefield et al., 2021) has shown that surface depressions may not reliably indicate subglacial lake extents due to the influence of viscous ice flow, and this should be addressed more specifically. Maybe it is simply a question of the terminology, so that you do not claim to delineate subglacial lake perimeters but rather subglacial lake expressions or signals?	128	Done. The subglacial lake “perimeter” / “boundary” terminology used in the manuscript is used to define the surface expression of this feature, so we thank the reviewer for highlighting the potential for misunderstanding here. We agree that there are factors, including viscous ice flow (Stubblefield et al., 2021) that might cause the surface expression of the subglacial Lake (SGL) to not fully reflect water movement at the ice base. To address the comment we have moved the following sentence, from section 2.3, to the first mention of delineating lake boundaries in section 2.2. We have acknowledged the point also in the discussion. We hope this clarifies that our ‘boundary’ terminology relates to the surface expression of a feature at the ice base that we do not directly observe. We have also added in the reviewers recommended citation, to highlight an example of the processes that might prevent this assumption from holding: “We note that processes such as the viscous deformation of ice or the impact of ice flow may alter the surface impression of behaviour at the ice bed (Stubblefield et al., 2021).” <line 156> We add the following here, for clarity: “The use of “subglacial lake boundaries” in this manuscript refers to the surface impression of elevation change observed due to subglacial lake filling and draining, rather than a direct delineation of the physical subglacial water edge. In order to measure the maximum extent of ice elevation change over each subglacial lake, (...)” <line 157> We also add to the discussion: “Our subglacial lake boundaries and volume estimates also provide new case studies for investigating the correspondence between basal and

		surface dynamics and the effect of ice flow on subglacial signals observed from the surface.” <line 904>
1.3	Additionally, the volume estimates are derived from a simple equation (Eq. 3) that lacks detailed justification, and it is unclear if this method meets current standards in the field (Stubblefield et al., 2021).	Done. We note reviewer 1’s concerns with the volume estimate calculation. However, reviewer 2 has conversely stated that they value having these numbers presented in the manuscript. Given the difference in opinion we have decided to retain the volume estimate calculations as we do think they will be of interest to the community. We have also ensured that Equation 3 clearly states the method used to calculate this number. We also now include the following two sentences in section 2.3: “We note that equating these two volumes varies in accuracy as a function of ice thickness, subglacial lake drainage rate and basal drag coefficient (Stubblefield et al., 2021). We therefore perform a simple volume calculation (Equation 3) and suggest coupled ice-hydrology models for the new subglacial lake networks presented in this study for future work, in order to assess variability in the surface-bed relationship.” <line 195> To reflect the reviewer’s concern, we have taken care to not state the volume estimate numbers in the abstract or conclusions, therefore giving them slightly lower prominence in the manuscript. We have also only included volume estimates, both in Table 1 and Supplementary Table 1, where we have lake boundaries delineated from surface elevation change maps produced from a 2-year period of surface elevation change measurements. Where boundary delineation does not allow us to calculate an area within this 2-year period, due to data coverage or the short time frame for some drainage/filling events, we mark this with a “-“ in Table 1, and a blank space in the Supplementary Table.
1.4	The study presents a significant expansion of the known inventory of active subglacial lakes in Antarctica and it offers a rare, decade-long dataset of lake activity. This contributes to our understanding of Antarctic subglacial hydrology, which is a topic that is important and timely.	Thank you for your comments on the significance and timeliness of our study.
1.5	However, the significance of the volume estimates is less clear, especially given uncertainties in methodology. The most impactful findings relate to the detection, variability, and spatial distribution of the lakes rather than their calculated volumes. Therefore, I would recommend that the authors reconsider if the lake volume estimation actually adds anything crucial – and if not, that they take that part of the study.	Done. We have addressed reviewer 1’s concerns about the volume change numbers in answer to comment 1.3 above. Please see this comment.

1.6	Impact: I would like to see the authors elaborate on the impact of their study. How exactly will our understanding be improved by the detection of these new active subglacial lakes? I see one obvious one, being the fact that basal melt is currently neglected in mass balance studies of the AIS. And as you state in L 30-31 the SGL variability can inform us on the water availability at the base.	Done. Thank you, this is a great point, and we have provided more detail in discussion section 4.1 of the paper: “Combining our new subglacial lakes with those documented in existing inventories provides a more comprehensive picture of subglacial lake distribution in Antarctica (Fig. 1). This is important as subglacial lake dynamics are currently not accounted for in ice mass loss projections, yet subglacial discharge can have significant impacts on ocean melting of ice shelves, ice stream thinning and acceleration (Gourmelen et al., 2025). Furthermore, subglacial lake variability detected in this study can inform us on water availability at the ice base, projections of which have thus far been limited due to a scarcity of observations.” <line 581>
1.7	The use of CryoSat-2 radar altimetry to track surface elevation changes is appropriate and well established. The spatial and temporal resolution is very suitable for the research objectives. However, I find that parts of the methodology needs improved explanation. The approach to assessing uncertainty in subglacial lake delineations (manual drawing of a few lake perimeters is simplistic and I believe it does not fully address an important source of uncertainty: the mismatch between surface signal and actual lake geometry. This needs to be made clearer throughout the manuscript. Further with respect to the delineation; outlines are provided less frequent than the elevation change results. How come? I would think that the outlines would change continuously and that more regular outlines (and hence areas) are needed for the volume calculation. Here, I am especially thinking of the outline change caused by ice deformation.	We thank reviewer 1 for this positive assessment of the dataset chosen for our research objectives. Done. We have addressed the reviewer’s concerns regarding surface signal vs. subglacial water geometry in 1.3 above. Please see this comment. Comment. The outlines provided are less frequent because, as stated in the methods, “We do not provide volume change estimates for 79 cases (53%) where the subglacial lake boundary was not delineated for a discrete period of activity. This usually corresponds to more rapid (less than 2 years) filling or drainage events where poorer altimetry data coverage leads to higher noise levels in the surface elevation change maps, making it hard to delineate the lake boundary.” As stated by reviewer 1, subglacial water areas do change continuously. However, indirect signals of this at the ice surface are only able to be delineated over longer time periods due to measurement noise in swath data for periods shorter than 2 years. For this reason, we implemented a minimum threshold of 2 years of data in the methods section (line 123): “We solve Eq. 1 within each grid cell using an iterative least-squares fit (...) based upon the following set of statistical thresholds: (...), a time series length of at least 2 years...” Furthermore, we investigate the magnitude of lake signal detectable with our processed CryoSat-2 data in response to reviewer 2’s comment below. Please see comment 2.14 for more details.

1.8	When removing outliers (L. 88) a threshold is applied on how far from a reference DEM the altimetry data can be to be included. The threshold is 25 m. Does this mean that you will not be able to detect surface depressions deeper than 25m? Does this not potentially filter out some subglacial lake variability signals?	88	Comment. This threshold is increased from that used in previous published work (Slater et al. (2021) use a 10 m/yr threshold), so is in line with methods best practice for surface elevation change studies. We increase this threshold from 10 m/yr to 25 m/yr to allow our method to capture “fast” subglacial lake surface elevation change signals, determined by reviewing existing literature. The threshold we have applied means that the elevation change rate would need to be more than 25 m/yr to be removed as an outlier, which would be much larger than any previously observed SGL drainage rate. To the best of our knowledge, the largest subglacial lake elevation change signal that has been observed is 20 m subsidence over Thwaites_124 subglacial lake, over an 18-month period, 2011-2013 (Smith et al., 2017). In our paper, we have not measured elevation change larger than 10 m on any individual lake, and this change happened over 5 years (on MoscowUniversity_19 subglacial lake).
1.9	Can you be more specific about what the standard error of the height change is? How is that derived? (L. 101).	101	Done. We have edited the text to be more specific regarding the standard error (ie. Stdev/\sqrt{n}) of the modelled elevation anomalies in each grid cell. “The elevation anomalies are averaged over 90-day epochs to produce mean elevation change time series within subglacial lake boundaries and their surrounding area. We estimate the uncertainty in our regional elevation time series at each epoch by computing the average of the standard error (σ/\sqrt{n}) of the model elevation anomalies within all contributing grid cells (Shepherd et al., 2019, Slater et al., 2021).” <lines 139>
1.10	The analytical approach is generally sound in terms of identifying elevation change events and characterising their timing and extent. However, there are a few things that could be improved: Please elaborate on the correlation described in Eq. 2. It is not clear to me why you would expect such a relationship in swath-processed altimetry data. The Davis et al., (2004) reference is for a study using conventional RA from ERS-2, using a retracking algorithm that can be sensitive to correlations between the waveform and estimated height.		Done. Power corrections are now routinely applied to all radar altimetry surface elevation change studies (McMillan et al., 2014; Gourmelen et al., 2018), so we have applied this correction to stay in line with that. The purpose of this study was to apply existing SEC methods to a new time period and region to detect new SGL activity, so it was out of scope for us to change the surface elevation processing chain substantially here. We have updated the text and citation for clarity to address this comment. “We account for spurious fluctuations in range associated with changes in radar echo shape by applying an empirical correction based upon correlated changes in elevation and backscattered power, consistent with existing CryoSat-2 SARIn altimetry studies (McMillan et al., 2014; Slater et al., 2021). To do this, we compute the correlation gradient in elevation as a function of power, dz/dp, using a linear fit in each grid cell. The change in backscattered power dp is multiplied by this correlation gradient and then subtracted from the

			model elevation anomalies to produce power-corrected anomalies (McMillan et al., 2014): *Equation 2* .” <line 134>
1.11	As mentioned above, the volume estimation via Eq. 3 requires additional justification and comparison with more advanced techniques. Reference to recent work such as Stubblefield et al. (2021) is warranted and should be included to contextualize the limitations of the current approach.		Done. Please see our response to Reviewer 1 comment 1.3 for a comprehensive discussion of the volume change estimations.
1.12	The current title, "Detection and variability of 85 new active Antarctic subglacial lakes with CryoSat-2 radar altimetry, 2010–2020", appears somewhat awkward in phrasing to me. Particularly the construction "variability ... with" may suggest that variability is being performed using the instrument, which is conceptually unclear. I recommend rephrasing for clarity and grammatical correctness. For example: "Detection of variability of 85 new active Antarctic subglacial lakes from CryoSat-2 radar altimetry (2010–2020)".	1	Done. We thank reviewer 1 for highlighting the grammar in our title. In response, we have edited the title and removed punctuation to align with the Nature Communications guidelines: "Detection of 85 new active subglacial lakes in Antarctica from a decade of CryoSat-2 radar altimetry data"
1.13	What is meant by worldwide here? You are only discussing subglacial lakes in Antarctica.	14	Comment. Worldwide means across the world. Active subglacial lakes have been found under the Greenland Ice sheet (Bowling et al., 2019) and under some glaciers and ice caps (Björnsson, 2003). As the statistic doesn't relate to only Antarctica, we have stated the larger relevant geographical region.
1.14	Do you mean complete cycle or complete drainage?	16	Done. We define a complete cycle as trough to trough or peak to peak, i.e. both filling and draining combined. A complete drain, or complete fill, would be only half a cycle. We define these terms in line 63 of the paper. "Of the 779 subglacial lakes identified globally, 681 are located in Antarctica, 20 % (146) of which have exhibited surface elevation changes suggestive of lake draining and filling cycles ^{17,18} . An observed cycle refers to a subglacial lake filling event along with its subsequent drainage event. "Active" subglacial lakes, which exhibit these cycles, were first observed (...)" It was not possible to include these definitions in the abstract of this paper for length constraint reasons, so we hope it is helpful to clarify that the terms are fully defined early in the paper.
1.15	At this stage it is not clear what a cluster or a network refers to	17	Done. We have updated the text where we define clusters and networks. " ... five new subglacial lake networks, with concurrent upstream lake draining and downstream filling, and 25 lake cluster located within a small geographical region, improving our knowledge of..." <line 18>
1.16	I would change valleys to depressions.	22	Comment. We use the word "subsidence" consistently in the manuscript to describe surface elevation change due to subglacial lake drainage. Therefore, we have decided to keep this wording as

			“subglacial valleys” here, to avoid confusion for the reader between small surface changes vs. relatively large magnitude variability in bedrock topography.
1.17	Provide exact number instead of percentages.	37	Done. We have added the absolute numbers in brackets next to the percentage as requested, so readers can read the metric they prefer. “Of the 779 subglacial lakes identified globally, 681 are located in Antarctica, 20 % (146) of which have exhibited surface elevation changes suggestive of lake draining and filling cycles.” <line 61>
1.18	Equation 1: I believe that you have not explained all the terms in the Eq.	E1	Done. We have updated this text to ensure all terms are described. “We model the elevation z as a function of local surface terrain (x, y), time (t), satellite heading (h) and the mean height within each grid cell (\bar{z}): $z(x, y, t, h) = \bar{z} + a_0x + a_1y + a_2x^2 + a_3y^2 + a_4xy + a_5h + a_6t \quad (1)$ The satellite heading (h) is a binary term set to 0 or 1, based on whether the satellite track is ascending or descending, respectively. We solve Eq. 1 within (...) <Line 118>
1.19	I would suggest to not provide the average time periods as I do not see how these numbers provide any useful insight, since these lakes are located in different regions and have different driving processes.	174-175	Done. The time-period of subglacial lake drainage events changes their effect on ice dynamics. When this water reaches the grounding line, subglacial lake drainages sustained over a long time-period (several years) can increase ice shelf basal melt rates and contribute to ice stream thinning and acceleration to a much greater extent than transient pulses of subglacial water from faster lake drainages which take months (Gourmelen et al., 2025). “The median time-period for a complete subglacial lake drainage event in this study is 2.2 years, with the median time for lake recharge being 3.5 years.” <line 265> We thank the reviewer for their comment, as it has prompted us to validate the inclusion of these statistics by adding the following point to our discussion: “Subglacial lake drainage events observed in this study sustained over relatively long time periods (a median subglacial lake drainage time of 2.2 years and an upper quartile of 4.4 years), which suggests that a large percentage of these subglacial lakes have potential to affect ice dynamics (Gourmelen et al. 2025).” <line 578>
1.20	Table 1: Please include uncertainties	T1	Done. We have added uncertainties to Table 1.
1.21	Fig 2. The scale number for Mill 165 is partly hidden behind the figure.	F1	Done.

Reviewer 2:

ID	Comment	Line	Response
2.1	It is my view that this study is of broad interest to the cryospheric community as it builds upon previous work focused on subglacial lake detection beneath the Antarctic Ice Sheet, especially in the context of subglacial hydrology and ice-sheet dynamics, by extending the record of active subglacial lake activity and their time-evolving filling and draining behaviour. I think the ice sheet modelling community would be very interested to use this dataset as validation at an ice-sheet scale in future. Overall, this is a well written manuscript which could be clearer some places, detailed by my specific comments. Once the authors address these, I can therefore recommend that this manuscript is suitable for publication in Nature Communications.		We thank the reviewer for their positive comments, time, attention to detail and constructive feedback.
2.2	In the abstract and several other places in the manuscript, 5 new subglacial lake networks and 25 clusters are mentioned. However, it is unclear to me what these 25 clusters refer to – are these lakes that intersect with modelled subglacial water flux (LeBrocq et al., 2013)? From Figure 1, it is unclear which of the 85 new lakes constitute these 25 clusters, and it is difficult to see in any detail the relationship between the authors' new reported active lakes and modelled subglacial water flux. This latter point is not really elaborated on in the manuscript.		Done. Please see reviewer comment 1.15 for updates on subglacial lake network and cluster definitions. We have also updated the ocean area in Fig. 1 so that the ice sheet is surrounded by a Southern Ocean bathymetry dataset, in similar colours to that of the subglacial hydrological pathways. We hope that this, along with increasing the opacity of the subglacial pathways in the figure, will help the reader identify and follow these pathways between subglacial lakes more easily. Inclusion of this dataset helps to demonstrate that active lakes often occur on modelled pathways, which we elaborate on in Figure 4.
2.3	The manuscript does a great job of validating lake outlines using previously reported lake derived from other sensors. I would like to suggest that the results of our recent study (Arthur et al., 2025) would be a valuable addition to this comparison, as it provides one of a few records of active subglacial lakes near the grounding line with several lakes coincident or located near those identified in this study. Specifically, locations such as Lazarevisen (coincident with our Lake L1), Vigridisen (46 km upstream of our lake V1), Muninisen (170 km upstream of our lakes M1 and M2) and Roi Baudouin (78 km downstream of our Lake R2) could provide useful points of comparison. I have included two of these examples below. In particular, relevant discussion could be added to the intercomparison of subglacial lake inventories (Section 4.1) and in Supplementary Figure 3.		Comment. We thank reviewer 2 for their positive assessments of our method validation. Whilst we agree that incorporating recent results from Arthur et al., 2025 would be a great addition to this work, we did not see significant surface elevation change signals at these locations in our 10-year elevation change map from which we selected our points of interest for this study. We recognize that Arthur et al. use predominantly ICESat (2003–2009) and ICESat-2 (2019–2023) data for their study, which may explain why we only observed one of the same lakes in our 10-year elevation change map, as our study periods only overlapped from late-2019 to mid-2020. Reprocessing our data for this area would be suitable for a more in-depth case study of this region, possibly amalgamating CryoSat-2 observations with ICESat missions, to fill in the 2009–2019 gap at a higher time resolution than possible with the REMA DEM strips used in Arthur et al. We think this would be really exciting future work. Extending the validation to cover more lakes is unfortunately outside of the scope of this manuscript. We have updated our paper to cite the reviewer recommended manuscript in the following places:

		“When subglacial lake drainage events occur and water discharges into the ocean, the flux of water can drive high, localised regions of basal melt on ice shelves through turbulent mixing from ocean plumes^{14,54}. Basal meltwater discharge downstream from subglacial lakes across the grounding line, resulting in these processes, is potentially more likely to occur at grounding line proximal subglacial lakes, such as that observed 15 km upstream of Muninisen Ice Shelf in Dronning Maud Land (Arthur et al., 2015). Given that 11 of our new subglacial lakes are closer than 15 km to the grounding line, these processing may be influencing ice melt even more than the limited recent observations.” <line 613>
2.4	In the Discussion (Section 4.3), I suggest mentioning the lake cluster upstream of Jutulstraumen (Jutulstraumen_69, _71, _74), which you detect filling and draining between 2013–2015 and 2017–2020. This cluster is located 81 km upstream of the active lake cluster reported by Neckel et al. (2021) using Sentinel-1 DInSAR, validated with ICESat-2, and observed to fill and drain between 2017–2020 (see right-hand map). Given their proximity and temporal activity, these lakes may form part of a larger interconnected network and could be associated with cascading drainage.	Done. We thank the reviewer for this suggestion. Discussing the proximity of the Jutulstraumen cluster we have observed here (Jutulstraumen_69, _71, _74) to the network upstream reported by Neckel et al. (2021) highlights a key finding in the paper. We have added this to the discussion as suggested. “Furthermore, the aforementioned cluster at Jutulstraumen Glacier is 81 km downstream of an active subglacial lake cluster observed to fill and drain between 2017–2020¹⁸. Given their proximity and temporal activity, these lakes may form part of a larger interconnected network and could be associated with cascading drainage.” <line 684>
2.5	One of the interesting findings from this new inventory is the detection of active lakes near the grounding line in certain locations (e.g. King Baudouin East_3, less than 4 km from the GL). I think this deserves further emphasis (for example, an additional figure highlighting some of these new examples) due to its important implications for subglacial water routing and the impact on local ice dynamics at the transition between grounded and floating ice in these regions.	Comment. This is a brilliant idea, but we felt this was a whole additional block of work which would be better served by a more in-depth additional paper. This would allow us to look at localized impact of lake drainage on short term grounding line migration and basal melt rates where there are ice shelves. Our assessment was that this was a huge amount of extra content that would be best presented in a stand-alone paper. As we are freely publishing our lake datasets with this paper, it will enable the scientific community to do this work if we are not quick enough ourselves.
2.6	As a general point, some of the discussion about future work suggestions could be more suited to Section 4.5 (Future Outlook), for example the last few sentences of Section 4.1 and 4.2.	Done. We moved: “While the likely routing of subglacial hydrological pathways has been modelled^{15,55}, very little is known about the temporal variability of this water flux. Subglacial lake activity provides one of the few mechanisms for measuring subglacial hydrology variability with any frequency across the whole ice sheet, therefore, we hope these results will be used inform future work in this area, for example, as validation dataset for high resolution subglacial water routing models.” From 4.1 to the future outlook section. <Line 892> We also edited and moved:

			“Future satellite missions and studies should therefore extend the record of subglacial lake activity on all documented lakes, in addition to exploring locations where additional subglacial lake networks may exist, by assessing both short and long-term ice sheet surface elevation change signals.” From 4.2 to the future outlook section. <line 832>
2.7	It is good to see details of the subglacial lakes, drainage/filling periods and associated area and volume changes provided in the Supplementary Data. I would encourage the authors to consider making their code available to reproduce the results presented in this manuscript, in line with Nature Communication guidelines and to promote reproducibility.		Done. The altimetry processing chain is taken from existing studies, therefore we do not provide the code here as it is available elsewhere (Slater et al. 2021). We have added elevation time series data and code to reproduce these figures to the data repository in response to this comment, to make the article of further use to the community. “The CryoSat-2 satellite altimetry data are freely available from the European Space Agency (https://cs2eo.org/), and we provide time series of elevation change over all 85 new subglacial lakes in our data repository. Code availability. The above-referenced repository contains code to reproduce elevation change time series over each new subglacial lake presented in our study. The altimetry processing chain is available from Slater and others (2021) ³⁶.” <line 940>
2.8	The Abstract refers to 25 clusters, and it is nice that you show individual lakes in Supplementary Figure 2, but it is also difficult to see how these relate to each other. Might be nice to have an additional figure highlighting the clusters and the relationships between the lakes.		Done. We have produced a new figure 4 that illustrates all 5 networks. <line 730>
2.9	I would specific ‘lake clusters’ here.	17	Done.
2.10	I’m not sure that we can definitively say that ice thicknesses prevent surface meltwater from reaching the bed, given that direct observations of this are currently lacking here. I think you also meant for this sentence to join with the next one?	26	Done. Thank you, we have updated the text. “In Greenland, subglacial lakes can form when seasonal surface melt percolates down from the ice surface to bedrock via moulins, crevasses ⁵ and hydrofractures ^{6,7}. Beyond the Antarctic Peninsula ⁹, there is limited evidence that surface water reaches the bed in this way in Antarctica. Antarctic subglacial water is primarily produced (...)” <line 24>
2.11	I suggest rewording this to ‘Our understanding of subglacial melt rate magnitudes’ rather than what they are.	29	Done.
2.12	Which inaccessible region are you referring to here – Antarctica as a whole?	30	Done. “inaccessible region of the ice base” <line 29>
2.13	There is some inconsistent use of the terms ‘boundaries’ vs ‘perimeters’ when referring to lake outlines, so please choose one and amend accordingly.	69	Done. All instances of “perimeters” changed to “boundaries”.
2.14	Can you specify the magnitude of lake signal detectable with processed CryoSat-2 data (how small)?	85	Done. In principle radar altimeters can measure surface elevation change with centimeter precision (Shepherd et al., 2019, Slater et al., 2021). We calculated an error estimate on the surface elevation change measured in this study and show that it is on

			average +/- 0.1 m (Table 1), which is in line with previous studies. This suggests that the methods we have applied to satellite radar altimeter data would not enable us to confidently detect SGL activity below this level. Other methods of detecting SGL activity, including laser altimetry and in situ GPS data do have higher vertical resolution on their measurement accuracy, but these datasets don't always have the spatial and temporal coverage required to monitor unpredictable SGL filling and draining continuously. Hence, long term continuous monitoring provided by CryoSat-2 was required in order for the satellite to be collecting data at the time of the activity. In addition, we implemented the following thresholds in our processing, which are detailed in the manuscript text: "To maximise the likelihood of ice surface height change reflecting real subglacial lake activity we pinpoint areas of spatially consistent surface height change ≥ 5 km that are distinct from measurement noise." <line 146> "We draw the lake contour at the ± 0.3 m yr-1 elevation change threshold and we adjust this depending on the signal magnitude, altimetry data coverage, and elevation patterns compared to the surrounding ice." <line 176>
2.15	I would repeat the observation period here to help the reader: 'We use our 10-year elevation change map (Oct 2010 – July 2020) to identify (...)'. (...)'.	105	Done.
2.16	I suggest 'with a major axis' is slightly confusing and not needed – consider changing the sentence to 'areas of spatially consistent surface height change ≥ 5 km that were distinct from measurement noise'.	107	Done.
2.17	Was the ± 0.3 m yr-1 elevation change threshold selected based on previous approaches, or based on judgement after trying different thresholds or following a different approach?	122	Done. We sensitivity tested different thresholds for delineating lakes and found that 0.3 m yr-1 was the minimum level where we could generally see a lake signal distinct from noise. We found that previous studies often didn't report this type of information in the methods sections, so hopefully its helpful for future studies that we have reported our methods with this level of detail.
2.18	I suggest rewording caption to 'Existing subglacial lake inventory is also shown'. I also think it would help the reader by labelling key regions as mentioned in the text, e.g. Wilkes Land, Victoria Land, Adelie and Bakutis Coasts, as well as major ice streams. Subglacial water flux should really be included in the Figure legend.	F1	Done.
2.19	I think it would be useful to refer to Figure 2 earlier in this paragraph, for example here.	172	Done. Sentence should have referenced figures 2l and 2k instead of 3l and 3k. Rectified.
2.20	Specify that the median lake drainage volume is previously-reported.	178	Done. "(...)an order of magnitude larger than the previously reported median Antarctic subglacial lake drainage volume of 0.12 km ³ ³⁹ ." <line 269>

2.21	Should Panel e (Moscow University 44) include a dashed line to indicate when it starts to fill? Also caption should read (...) compared to elevation change from non-subglacial lake regions (...)?	F2	Done. Yes, thank you for spotting this. Dash added to Panel e for when the subglacial lake starts to fill (2012). “compared to elevation change from non-subglacial lake regions (light green lines)” <line 288>
2.22	Refer to Fig. 2l here.	189	Done.
2.23	The wording here is slightly confusing, I suggest changing to: ‘where we observe quiescent periods during filling or draining’.	196	Done.
2.24	Unclear to me why some lake volume change estimates are starred, please explain in caption.	T1	Done. We thank the reviewer for pointing this out. Starred volume estimates did represent volumes estimated from a lake boundary at least +/- 2 years different to the period of filling or draining the estimate was calculated for. As these numbers were here for illustrative purposes and not included in Supplementary Table 1, due to reduced accuracy, we have redacted them from Table 1 and explained this in the caption. “Blank volume estimates could not be accurately estimated, due to lake boundaries at least 2 years different to the period of filling or draining.” <line 411>
2.25	The meaning of this sentence is unclear, given the previous sentence.	229	Done. We thank reviewer 2 for their comment on this sentence. Upon further reflection, we have revised this section of work to make the results clearer. “The resulting time series of subglacial lake outlines shows that the area of subglacial lake activity-induced surface elevation change can evolve by up to 50 %, in the case of two filling events at Scott_12 subglacial lake (Fig. 2k, Table 1, Supplementary Table 1) where the area of 59.4 km ² for the 2014–2016 filling event decreases to 29.8 km ² for the later 2018–2020 filling event. The minimum amount of area change we observe between cycles is 9 % , for 2 drainage events at Lambert_84 subglacial lake (Supplementary Table 1, Supplementary Fig. 2), indicating large variability on individual lakes.” <line 420> Sentence deleted: “Moreover, our results show that only three subglacial lakes, Cook West_67, Institute_14 and Mill_161, fill and drain over the same area constantly (within < 1 km ² change between active periods) (Supplementary Table 1).”
2.26	Suggest specifying how much area change.	235	Done. Please see our response to reviewer comment 2.25 above.
2.27	Suggest adding small insets to each panel to help situate lake locations in Antarctica.	F3	Done.
2.28	Suggest it could be clearer to refer to ‘activity rate patterns’ as modes of activity, here and on L352.	350 + 352	Done.
2.29	It seems unclear to me how you arrived at this point, as I can’t find any mention of this or any	432	Done. We thank the reviewer for this comment and think that on reflection we need to soften our

	correlation analysis with these variables. I see that ice speed and ice thickness for each lake are reported as part the supplementary dataset, although mean drainage flux is not. Could you elaborate further?		language here. We did a preliminary assessment of the statistical correlations between the lake activity on our new lakes and a range of geophysical parameters and didn't find a clear correlation between them, hence our statement in the paper. However, in reality, this is a topic that needs further work, running the correlations over all known SGL's rather than only our new lakes, and investigation into the implications of any findings in more detail. We have changed the text to reflect this. "Large variability in both the magnitude and frequency of subglacial lake activity reflects the heterogeneity of their geographical and geometric settings in Antarctica, including factors such as ice thickness and flow speed local to the subglacial lake. Despite evidence that basal hydrology can be highly sensitive to these geophysical parameters⁵⁶, an initial assessment suggests that there is no evidence of a correlation with either the four distinct behaviours listed above or mean lake activity, but in the future, further work is required to assess this over all Antarctic subglacial lakes." <line 623> '..., but in the future, further work is required to assess this over all Antarctic subglacial lakes.' <line 654>
2.30	Very long sentence, suggest splitting.	360	Done. "As complete subglacial lake fill-drain cycles typically last about a decade in Antarctica, it is possible that our 20-year record of ice sheet wide surface observations is currently too short to assess whether subglacial lakes change their modes of behaviour over time." <line 654>
2.31	This is mentioned as a general point above, but it is unclear what the 25 clusters are that are referred to, and what you mean by 'the same' modelled subglacial hydrological pathways.	368	Done. Please see reviewer comment 1.15 for updates on subglacial lake network and cluster definitions. Sentence altered to the following for clarity: "Of 25 lake clusters situated along the same modelled subglacial hydrological pathways¹⁵, indicating hydrological connectivity, our results show that five study areas exhibit upstream drainage events concurrent with downstream filling." <line 662>
2.32	I suggest rewording the term 'length-scale' to, for example: 'indicates decadal variability in connected subglacial system'.	383	Done. "Eight years of potential interconnectivity observed at the Français Glacier subglacial lakes, (...), indicates decadal variability in connected subglacial systems." <line 656>
2.33	It would be relevant to cite Neckel et al. (2021) here.	412	Done.
2.34	By bed properties, are you referring to sediment saturation or other parameters?	434	Done. We have updated the text to clarify this. "bed properties, such as geometry, sediment saturation and temperature, or the volume of subglacial water available." <line 811>

2.35	cycles and activity to me refer to similar things, so I would reword this to 'subglacial lake filling and draining'.	444	Done. "filling and draining patterns" <line 820>
2.36	Suggest changing 'pooling' to 'ponding'. Similarly, suggest changing 'pools' to 'lakes' on L466.	462 +	Done. 466
2.37	Suggest changing 'activity variability' to 'filling and draining activity' for clarity.	464	Done.
2.38	no need to hyphenate 'subglacial'.	17	Done.
2.39	Add comma after Antarctica.	25	Done.
2.40	The methods' reliance (add apostrophe).	50	Done.
2.41	New sentence: 'However, this is limited by (...)'. <line 76>	51	Done. "However, this is limited by" <line 76>
2.42	New sentence: 'However, these measurements (...)'. <line 76>	53	Done.
2.43	No need to hyphenate 10-years.	68	Done.
2.44	No need to capitalise ice sheet.	79	Done.
2.45	New sentence: 'In the event that the map is produced over the full period of the cycle, (...)'. <line 76>	119	Done.
2.46	Add 'measurements' after 'subglacial lake area'. <line 76>	136	Done.
2.47	New sentence: 'Therefore, we assume (...)'. <line 76>	137	Done.
2.48	New sentence: 'However, we note that (...)'. This sentence would also benefit from a citation. <line 76>	142	Done. Change to new paragraph, so "however" redacted. Citation added as suggested. "We note that (...) (Stubblefield et al., 2021)." <line 156>
2.49	For brevity, I suggest rewording to: 'We identify 85 new active Antarctic subglacial lakes (Fig. 1)..'. <line 76>	150	Done.
2.50	37, not thirty-seven. Also suggest bracketing (subglacial lakes Whillans_180 and Scott_12). <line 76>	172	Done. "During our study period, 37 subglacial lake drainage events and 34 filling events are captured in their entirety" <line 263>
2.51	New sentence: 'This lake drained approximately 1.3 km ³ (...)'. <line 76>	177	Done.
2.52	Suggest slight rewording for clarity: 'for example Totten_52, which was associated with 8 m of uplift'. <line 76>	194	Done. "which was associated with 8 m of both subsidence and subsequent uplift" <line 300>
2.53	Specify 'subglacial lake filling and draining episodes' here. <line 76>	201	Done.
2.54	Missing full stop. New sentence following line: 'Therefore, elevation change (...)'. <line 76>	260	Done.
2.55	Hyphenate 'decade long'. <line 76>	263	Done.
2.56	'Adjacent to, but distinct from, the main Institute E1 lake area'. I think the following sentence is also unnecessary and could be deleted. <line 76>	301	Done. Done. (Sentence deleted).
2.57	Given the point made in the previous sentence, I suggest this sentence is unnecessary. <line 76>	307	Done. (Sentence deleted, see above, comment 2.56).
2.58	Suggest this sentence can be more concise: 'Previous subglacial lake inventories document the existence of active subglacial lakes beneath fast ice streams alongside less active lakes towards the interior of the continent, often beneath slower flowing ice1, 11, 40, 41, 45, 47, 48'. <line 76>	314	Done. "Previous subglacial lake inventories document the existence of active subglacial lakes beneath fast ice streams alongside less active lakes towards the interior of the continent, often beneath slower flowing ice1, 11, 39, 40, 45, 47, 48" <line 567>

2.59	Suggest 'subglacial lake clusters in East Antarctica.' (...) Secondly, our results show (...)'.	377	Done.
2.60	Change 'cascade' to cascading.	403 + 412 + 416	Done.
2.61	New sentence: 'Ice dynamics dominate mass loss in Antarctica, so it is important to quantify any impact from subglacial hydrology'.	429	Done.
2.62	New sentence: 'Therefore, subglacial lake systems (...)'.	439	Done.
2.63	New sentence: 'However, further observations are required so we can more accurately quantify the impact of drainage events on ice dynamic processes (...)'.	451	Done.
2.64	Comma, not semicolon.	454	Done.
2.65	New sentence: 'We provide the first time-evolving (...)'.	474	Done.

Reviewer 3:

ID	Comment	Line	Response
3.1	Even though, I am not an altimetry expert the methodology employed appears to be both robust and comprehensively delineated. Especially making use of the POCA processing approach allows a much finer spatial resolution for lake mapping. Therefore, this study presents a much more detailed and accurate dataset compared to the existing inventory as can be seen in Figure 3. It is positive to note that the shapefiles of the newly detected lakes have been published open access. The text is clear and accessible, with results presented in a well-structured manner.		We thank the reviewer for their time and positive assessment.
3.2	The significance of the work is given in any case due to the importance of Antarctica's subglacial hydrology for ice loss (Fricker et al. 2025). Nevertheless, for the scope of Nature Communications I would have expected more insights regarding the process understanding of subglacial lake activity and distribution. The authors define lake variability and behaviour into seven categories (L 351 ff.) and provide insights into connected subglacial networks (L 367ff.). But the data produced within this study would facilitate a greater level of insight with minimal effort. The following ideas could be pursued: - The supplementary includes all time series of the 85 analysed lakes. There seems to be a strong seasonality in the drainage and filling events. Did you have a look into the seasonality (e.g. detrending and performing a seasonality analysis on time series with sufficient		Done. We agree with the reviewer that the Fricker et al (2025) review highlights the importance of SGL's in the Antarctic systems. We believe that the size of our new dataset, which discovers 85 new active subglacial lakes and documents a previously unreported decade of filling and drainage events, will be of wide interest to the science community and the Nature Communications readership. We note that other reviewers commented on the suitability of this manuscript for this journal. Please see our response to reviewer comment 3.11 below regarding the potential seasonality of lake draining and filling events. We are excited that future studies will do exactly as Reviewer 3 suggests, rigorously investigating small amplitude signals and connecting filling and drainage events with physical processes, which we still know remarkably little about. It was our judgement that

	measurement points). Why is there strong variability in some years but not in other?		this work was substantial and would therefore need to be extended over all active Antarctic subglacial lakes, or all active lakes within a small geographical region, to be the topic of hopefully many future papers. We are making our lake boundaries and time series freely available to the community for this work to be done by both them and us. We have added in the reviewers excellent suggestions into the future work section (4.5) of the paper. “The significance of short-term variability on each subglacial lake elevation time series could be assessed by comparing magnitudes to localised snowfall variability.” <line 895>
3.3	- Subglacial lakes seem to be clustered in some specific areas (Figure 1). Can you explain why?		Please see reviewer comment 2.8.
3.4	- The lakes are categorized into seven categories. How is their spatial distribution? Have you tried to create a map based on these categories and where you able to identify any spatial patterns?		Done. We used categorisations determined from previous studies (Livingstone et al., 2019) in our analysis here, adding two categories which we observed in our dataset. We did not produce a map of these categorisations for this this study it was out of scope, however we have added in a comment to our future work section suggesting that this would be an excellent next step for a future paper. We suggest that this work could be done in a more comprehensive way by including our new lakes along with all the 146 previously identified active AIS subglacial lakes too, allowing spatial patterns to be identified. “By combining our observations of new Antarctic subglacial lakes with the existing inventory, a comprehensive analysis of the subglacial behaviour patterns expanded on in our study can be used to investigate spatial patterns of variability and the physical processes causing them.” <line 893>
3.5	Add the CryoSat data acquisition limit at the South Pole to the map to show where potential active lakes might have been detected.	F1	Done. We have added this shaded area to the figure legend and altered the formatting of this to make it stand out.
3.6	Can you provide any uncertainties for the volume change estimate? In L 157 you mention that the height change signal at the surface is attenuated by thick ice and hence the change in lake level is very likely different form the surface elevation change and might introduce large errors.	139 + T1	Done. We have added in error estimates for the elevation change data and included this new information into Table 1 for each lake. We have added a few sentences to the manuscript (please see reviewer comment 1.2) to reiterate that our data and methods only measure the surface expression of any lake filling and drainage event, we don't have any method of measuring how that corresponds to lake cavity area change beneath the ice. As ice thickness is up to 4km in some parts of Antarctica, we wanted to make the point that there is a possibility that the surface expression might not reflect the entire subglacial cavity area change. However, there is no data or method that we are aware of that has ever been able to investigate this

			point directly. Hopefully the readers will value having reasonable methodological limitations stated clearly. We have clarified the statement around this by adding in new detail, so hopefully this makes our statement even clearer.
3.7	What is the reason for your choice of these 13 lakes in particular?	T1	Done. These are the statistics for the lakes we have shown elevation change maps and time series for in Fig. 2. We chose to highlight lakes that had either particularly large drainage events, that had interesting repeat modes of filling and draining within our time period, or that had interesting other behaviour that would be of interest to the reader. We aimed to highlight the full range of behaviour that we observed, and we discuss all of these examples in more detail in the text. We note that our supplementary figures provide the same information for all lakes in the study, so if a reader had particular interest in a different lake to the ones we have highlighted, they would be able to look at this figure to get that information. “Table 1. Log of ice surface elevation change over subglacial lakes in Fig. 2.”
3.8	Also consider providing the median and quantiles for drainage events. Just the mean average could be miss leading in case of strongly varying time spans.	174	Done. With our study providing information on 85 new lakes, and data on 19 existing ones that were active during our time period, we have a large number of measurements and statistics to report. As such, in this paper, we had to be selective about the statistics we provided, so felt that the median was most useful. “The median time-period for a complete subglacial lake drainage event in this study is 2.2 years, with the median time for lake recharge being 3.5 years.” <line 265> We have provided the median and quartiles for the reviewer (see below) but felt that it would reduce readability of the sentence to provide this in this part of the main manuscript. We have now added some of this information to our discussion; please see reviewer comment 1.19. Subglacial lake drainage episode lengths Mean: 3.1 years Lower quartile: 1.2 years Median: 2.2 years Upper quartile: 4.4 years Subglacial lake filling episode lengths Mean: 4.1 years Lower quartile: 2.5 years Median: 3.5 years Upper quartile: 5.5 years

3.9	You mention that subglacial lakes can exist closer to the grounding line (GL) than previously thought. A recent study by Arthur et al. 2025 also mentions subglacial lakes close to the grounding line. Consider discussing these findings with yours.	328	Done. “When subglacial lake drainage events occur and water discharges into the ocean, the flux of water can drive high, localised regions of basal melt on ice shelves through turbulent mixing from ocean plumes^{13,53,56}. Basal meltwater discharge downstream from subglacial lakes across the grounding line, resulting in these processes, is potentially more likely to occur at grounding line proximal subglacial lakes, such as that observed 15 km upstream of Muninisen Ice Shelf in Dronning Maud Land (Arthur et al., 2015). Given that 11 of our new subglacial lakes are closer than 15 km to the grounding line, these processing may be influencing ice melt even more than the limited recent observations. Considering the enhanced ice melt and glacier retreat observed in the 21st century (...)” <line 617>
3.10	You used the MEaSURES grounding line. Did you compare the distance between the GL and your detected lakes with a newer grounding line product?	332	Comment. No, we used the MEaSURES grounding line throughout because we wanted to provide numbers from a widely used consistent product that would be easy to interpret. The timestamp on this dataset also maps on well to the period of our study. We think it could be interesting future work to investigate the ‘near-grounding line’ SGL’s in more detail, to learn more about the timing and impact of SGL water exit to the ocean, and the impact this might have on the glacier dynamics.
3.11	Here you present the drainage of Francais_46 and subsequent filling of Francais_36. Did you do any statistical analysis on the sub-annual fluctuations and the connection between drainage and filling? For example, in 2015 F_46 re-fills and drains half a year later. Do you see in return a filling signal in F_36 with a temporal lag?	F4	Done. We did not do a full statistical analysis on the shorter-term variations in these time series. We did a preliminary investigation to see if they might be seasonal, and we don’t think they are, but as we did not have the capacity to do an in-depth case study on every lake our results, we have not investigated this further. The reviewer makes a really interesting point about whether the shorter-term variability might be interlinked in addition to the long term (decadal) variability. We hadn’t considered this, but we think it would be an excellent topic for a future study, so have added it into our future work recommendations. We have included the below text in our future work section to highlight this comment: “The significance of short-term subglacial lake variability could also be assessed by comparing sub-annual elevation fluctuations on each subglacial lake to key weather variables, such as localised snowfall variability.” <line 900>
3.12	There seem to be some lakes that experienced a surface lowering but lake area expansion (e.g. Jutulstraument_71, Fisher_168). Is this a common pattern and what is the explanation behind?	Sup.	Done. Again, this is a fantastic question that we don’t think we can fully answer in this study. This paper reports for the first time that there are substantial lake area changes with subsequent filling and drainage events, and there are no previous studies even documenting that area change routinely occurs. Our study shows comprehensively that area change does happen to subglacial lakes, with 19 out of our 85 lakes having multiple drainage boundaries. We think that investigating this in detail on one lake will

			require a dedicated case study, or if we were to investigate this in a more systematic way over every lake, then our dataset would be a good start, but we may require a longer time series. Even our long dataset has only one filling and drainage event on most of the lakes (79 out of 85) so we would need to observe the same event multiple times in order to make more generalisable statements of this kind.
--	--	--	---

Additional edits made:

- All figure fonts changed to Helvetica to fit the Nature Communications requirements.
- Title added to all figure captions with bold/regular formatting updates where required.
- Abstract updated from 180 words to 149 in line with Nature Communications requirements.
- We have removed numbers from headings and subheadings.
- Typo updated: “The historical subglacial lake inventories suggest that a larger number of active lakes are located in West Antarctica compared to the East Antarctic Ice Sheet” <line 601>.
- Reference to Figure 1 made in Future outlook <line 834>
- “Competing interests. The authors declare no competing interests.” <line 959>

Response to referees: Detection and variability of 85 new active Antarctic subglacial lakes with CryoSat-2 radar altimetry, 2010–2020

Ref: NCOMMS-25-13112

Reviewer 1:

ID	Comment	Line	Response
1.2	The delineation of subglacial lake perimeter outlines from surface change signals is potentially overinterpreted. The assumption that surface expressions reflect the true subglacial lake boundaries is not adequately justified in the current version of the manuscript. Previous work (e.g., Stubblefield et al., 2021) has shown that surface depressions may not reliably indicate subglacial lake extents due to the influence of viscous ice flow, and this should be addressed more specifically. Maybe it is simply a question of the terminology, so that you do not claim to delineate subglacial lake perimeters but rather subglacial lake expressions or signals?	128	Done. The subglacial lake “perimeter” / “boundary” terminology used in the manuscript is used to define the surface expression of this feature, so we thank the reviewer for highlighting the potential for misunderstanding here. We agree that there are factors, including viscous ice flow (Stubblefield et al., 2021) that might cause the surface expression of the subglacial Lake (SGL) to not fully reflect water movement at the ice base. To address the comment we have moved the following sentence, from section 2.3, to the first mention of delineating lake boundaries in section 2.2. We have acknowledged the point also in the discussion. We hope this clarifies that our ‘boundary’ terminology relates to the surface expression of a feature at the ice base that we do not directly observe. We have also added in the reviewers recommended citation, to highlight an example of the processes that might prevent this assumption from holding: “We note that processes such as the viscous deformation of ice or the impact of ice flow may alter the surface impression of behaviour at the ice bed (Stubblefield et al., 2021).” <line 156> We add the following here, for clarity: “The use of “subglacial lake boundaries” in this manuscript refers to the surface impression of elevation change observed due to subglacial lake filling and draining, rather than a direct delineation of the physical subglacial water edge. In order to measure the maximum extent of ice elevation change over each subglacial lake, (...)” <line 157> We also add to the discussion: “Our subglacial lake boundaries and volume estimates also provide new case studies for investigating the correspondence between basal and surface dynamics and the effect of ice flow on subglacial signals observed from the surface.” <line 904>
	Thanks for the revisions. I acknowledge your explanation which is fair – but I would suggest that you rephrase the statement in the abstract to make it clear there as well.		Done. “We delineate time-varying boundaries of subglacial lake activity and investigate their variability over time.” <line 16>
1.8	When removing outliers (L. 88) a threshold is applied on how far from a reference DEM the	88	Comment. This threshold is increased from that used in previous published work (Slater et al. (2021) use a

	altimetry data can be to be included. The threshold is 25 m. Does this mean that you will not be able to detect surface depressions deeper than 25m? Does this not potentially filter out some subglacial lake variability signals?	10 m/yr threshold), so is in line with methods best practice for surface elevation change studies. We increase this threshold from 10 m/yr to 25 m/yr to allow our method to capture “fast” subglacial lake surface elevation change signals, determined by reviewing existing literature. The threshold we have applied means that the elevation change rate would need to be more than 25 m/yr to be removed as an outlier, which would be much larger than any previously observed SGL drainage rate. To the best of our knowledge, the largest subglacial lake elevation change signal that has been observed is 20 m subsidence over Thwaites_124 subglacial lake, over an 18-month period, 2011-2013 (Smith et al., 2017). In our paper, we have not measured elevation change larger than 10 m on any individual lake, and this change happened over 5 years (on MoscowUniversity_19 subglacial lake).
	I think that the threshold applied by Slater et al., (2021) is very appropriate given the aim of that study. But in this study the aim is different, and I do not understand why it is needed to have such a threshold. In McMillan et al., (2013) it is stated about the Cook_E2 lake that “Between November 2006 and March 2008, the ice surface fell by ~ 45 m” so isn’t that an example of a signal that you would miss?	Comment. Given the characteristics of the input radar altimetry data, we had to choose a threshold that provided a balance between noise reduction and subglacial lake signal identification. It is possible that our chosen threshold, while over double the Slater et al. threshold, is too small to capture the most extreme active subglacial lake signals, such as the example indicated on Cook. However, as the focus of this study was to provide a continent-wide assessment across the whole of Antarctica, our view was that a more conservative estimate would be appropriate for that context. In our initial look at the 10-year long surface elevation change map, we visually examined the raw and the thresholded data. This initial investigation, while not systematic, led us to conclude that our current method was appropriate for the study, as all the signals we wished to investigate from the 10-year elevation change map were visible with these thresholds applied. This initial investigation led us to conclude that our current method was appropriate for the study, as all the signals we wished to investigate from the 10-year elevation change map were visible with these thresholds applied. While our study is as complete and comprehensive as we can make it, it is of course possible we have missed an important SGL event that others will subsequently find.
1.10	The analytical approach is generally sound in terms of identifying elevation change events and characterising their timing and extent. However, there are a few things that could be improved: Please elaborate on the correlation described in Eq. 2. It is not clear to me why you would expect such a relationship in swath-processed altimetry data. The Davis et al., (2004) reference is for a study using conventional RA from ERS-2, using a retracking algorithm that can be sensitive to	Done. Power corrections are now routinely applied to all radar altimetry surface elevation change studies (McMillan et al., 2014; Gourmelen et al., 2018), so we have applied this correction to stay in line with that. The purpose of this study was to apply existing SEC methods to a new time period and region to detect new SGL activity, so it was out of scope for us to change the surface elevation processing chain substantially here. We have updated the text and citation for clarity to address this comment.

	correlations between the waveform and estimated height.	“We account for spurious fluctuations in range associated with changes in radar echo shape by applying an empirical correction based upon correlated changes in elevation and backscattered power, consistent with existing CryoSat-2 SARIn altimetry studies (McMillan et al., 2014; Slater et al., 2021). To do this, we compute the correlation gradient in elevation as a function of power, dz/dp, using a linear fit in each grid cell. The change in backscattered power dp is multiplied by this correlation gradient and then subtracted from the model elevation anomalies to produce power-corrected anomalies (McMillan et al., 2014): *Equation 2* .” <line 134>
	I have read now the three mentioned papers, and I apologize if I have misunderstood something, but I believe that neither Slater et al., (2021) or McMillan et al. (2014) used swath-processed data as you do here. Further, if I understood correctly, the power is indeed used in Gourmelen et al., (2018) but not for surface elevation change correction but for weighing individual elevation points from the swath, which is not what I understand that you are doing here. Please elaborate to justify this approach.	Comment. Its correct that Slater and McMillan use standard POCA radar altimetry data, but the Gourmelen et al., 2018 reference is a methods paper for processing swath mode altimetry data. All papers use radar altimetry data from the CryoSat-2 satellite, therefore use data at the same radar frequency which means penetration depth and broader interactions with the snowpack are the same. Overall, application of power corrections is standard practice, but as the reviewer points out there could be slight differences in the impact of the correction if applied at different stages of the data processing chain. The McMillan et al. power correction applied to retracked data, would likely not work as intended for swath processed data because the backscatter within a specific waveform has a wide amplitude with respect to range gate position, much more than the power variation due to scattering variability with time. At an earlier stage in the study we did run the analysis without the power correction to sensitivity test how much of an effect it made, but the impact appeared negligible. At an earlier stage in the study, we did run the analysis without the power correction to sensitivity test how much of an effect it made, but the impact appeared negligible. We retained the correction, because in addition to following best practice, we thought there were some cases where it seemed to moderately reduce the time series noise. Based on previous studies this is the type of benefit you would expect to see. As the focus of this study was to scientifically exploit mature methods and datasets for a new application, we didn't spend more time adjusting the core altimetry processing chain. This would be a good topic for future altimetry data intercomparisons, so we will feed that suggestion back into the relevant teams.

Reviewer 2:

ID	Comment	Line	Response
----	---------	------	----------

2.2	In the abstract and several other places in the manuscript, 5 new subglacial lake networks and 25 clusters are mentioned. However, it is unclear to me what these 25 clusters refer to – are these lakes that intersect with modelled subglacial water flux (LeBrocq et al., 2013)? From Figure 1, it is unclear which of the 85 new lakes constitute these 25 clusters, and it is difficult to see in any detail the relationship between the authors' new reported active lakes and modelled subglacial water flux. This latter point is not really elaborated on in the manuscript.		Done. Please see reviewer comment 1.15 for updates on subglacial lake network and cluster definitions. We have also updated the ocean area in Fig. 1 so that the ice sheet is surrounded by a Southern Ocean bathymetry dataset, in similar colours to that of the subglacial hydrological pathways. We hope that this, along with increasing the opacity of the subglacial pathways in the figure, will help the reader identify and follow these pathways between subglacial lakes more easily. Inclusion of this dataset helps to demonstrate that active lakes often occur on modelled pathways, which we elaborate on in Figure 4.
	In the Abstract, I appreciate the authors have tried to clarify their definition of what constitutes individual lake clusters, but this could be interpreted as 25 lake clusters located within (the same) small geographical region. I would therefore remove this from the abstract and instead add this to in the Discussion ('Connected Subglacial Lake Networks').		Done. Thank you for this comment on the readability of this point. We have removed this definition from the abstract, as we believe "cluster" is accurate enough to describe lakes which are clustered together. We leave the more detailed definition in the main manuscript for readers to clarify if needed. We have also changed the wording from "25 lake clusters" to "25 clusters of lakes" to make the description more clear.
	It may be worth adding a caveat that, while you observe that 25 lake clusters appear aligned along the same modelled subglacial hydrological pathways (LeBrocq et al., 2013), and this is used to infer hydrological connectivity, the underlying water routing methodology is based on the Bedmap2 grid. This grid can be significantly smoothed between radar profiles, meaning that flow routing is primarily influenced by topographic features near the closest radar tracks. As a result, for lakes located farther from these profiles, the inferred locations of subglacial streams may be overly confident or less reliable. For more info see Shackleton et al. (2023, https://doi.org/10.1029/2023JF007269).		Comment. We thank reviewer 2 for this comment and we fully agree with their concerns about gridding and over smoothing of the bed topography data based on measurement proximity. It's clear that the topography (well represented or not) is the primary driver of where water is expected to pond. We used the modelled subglacial hydrological pathway information only as a secondary dataset to infer possible connectivity between lake clusters, therefore, we think that including this caveat about the quality of the bed data would be slightly outside of the scope of the paper.
2.17	Was the ± 0.3 m yr ⁻¹ elevation change threshold selected based on previous approaches, or based on judgement after trying different thresholds or following a different approach?	122	Done. We sensitivity tested different thresholds for delineating lakes and found that 0.3 m yr⁻¹ was the minimum level where we could generally see a lake signal distinct from noise. We found that previous studies often didn't report this type of information in the methods sections, so hopefully its helpful for future studies that we have reported our methods with this level of detail.
	I appreciate the authors' explanation of their selection of the ± 0.3 m yr ⁻¹ elevation change threshold in their response to my comments, and I think it would be useful to include this in the main manuscript (L129).		Done. "We draw the lake contour at the ± 0.3 m yr⁻¹ elevation change threshold, based on noise sensitivity testing, and we adjust this depending on the signal magnitude, altimetry data coverage, and elevation patterns compared to the surrounding ice." <line 129>
2.66	I think it could be helpful for the reader to have a key in Figure 3 (as in Figure 4).		Done.

Reviewer 3:

ID	Comment	Line	Response
3.1	I appreciate the authors for their thorough responses and clarifications to all reviewer comments. It's creditable to see the manuscript enhanced in various aspects, particularly with the addition of Figure 4, which significantly contributes to the comprehension of lake clusters and their interconnections. I recognize that some of my questions related to lake dynamics necessitate further analysis that fall outside the scope of this paper. Therefore, I look forward to the authors' future publications that will likely explore the dynamics of subglacial lakes, including the potential seasonality in filling and drainage events.		We thank reviewer 3 for their time and positive assessment of our manuscript and revisions. We know how time consuming reviewing papers can be, so we are incredibly grateful for their generous support.

Detection and variability of 85 new active Antarctic subglacial lakes with CryoSat-2 radar altimetry, 2010-2020

Sally Wilson et al.

Jennifer Arthur (Referee) jennifer.arthur@npolar.no

General comments:

This manuscript uses CryoSat-2 radar altimetry swath-processed surface elevation data to develop a new dataset of active subglacial lake boundaries and time series of their filling and draining over a decade (2010-2020). The authors document complete filling and draining cycles and identify a number of interconnected subglacial lake networks. They found a total of 85 active subglacial lakes and five interconnected lake networks and conclude that individual lake areas can change by up to 280% over multiple periods of filling and draining.

Hydrologically active lakes are an important, yet poorly constrained component of the ice sheet subglacial hydrological system. Active lakes can transport excess meltwater episodically towards the grounding line, where it is discharged into sub-ice shelf cavities and can enhance ice shelf basal melting. Subglacial hydrology beneath the Antarctic Ice Sheet remains poorly understood and challenging to represent in ice-sheet models. Observing active lakes using repeat satellite data is therefore crucial to understanding the impact of subglacial hydrology on ice dynamics and the ice sheet-ocean system. Few studies have produced ice sheet-scale inventories of active subglacial lakes, and studies using different satellite sensors each have their own limitations, e.g. limited coverage.

It is my view that this study is of broad interest to the cryospheric community as it builds upon previous work focused on subglacial lake detection beneath the Antarctic Ice Sheet, especially in the context of subglacial hydrology and ice-sheet dynamics, by extending the record of active subglacial lake activity and their time-evolving filling and draining behaviour. I think the ice sheet modelling community would be very interested to use this dataset as validation at an ice-sheet scale in future. Overall, this is a well-written manuscript which could be clearer some places, detailed by my specific comments. Once the authors address these, I can therefore recommend that this manuscript is suitable for publication in *Nature Communications*.

I have a few general points, which I outline below:

- In the abstract and several other places in the manuscript, 5 new subglacial lake networks and 25 clusters are mentioned. However, it is unclear to me what these 25 clusters refer to – are these lakes that intersect with modelled subglacial water flux (LeBrocq et al., 2013)? From Figure 1, it is unclear which of the 85 new lakes constitute these 25 clusters, and it is difficult to see in any detail the relationship between the authors' new reported active lakes and modelled subglacial water flux. This latter point is not really elaborated on in the manuscript.
- The manuscript does a great job of validating lake outlines using previously reported lakes derived from other sensors. I would like to suggest that the results of our recent study (Arthur et al., 2025) would be a valuable addition to this comparison, as it provides one of a few records of active subglacial lakes near the grounding line with several lakes coincident or located near those identified in this study. Specifically, locations such as Lazarevisen (coincident with our Lake L1), Vigridisen (46 km upstream of our lake V1), Muninisen (170 km upstream of our lakes M1 and M2) and Roi Baudouin (78 km downstream of our Lake R2) could provide useful points

of comparison. I have included two of these examples below. In particular, relevant discussion could be added to the intercomparison of subglacial lake inventories (Section 4.1) and in Supplementary Figure 3.

Left: Lake L1 boundary in blue (Arthur et al., 2025) and Lazarev_35 in purple (this manuscript) upstream of Lazarev Ice Shelf. Right: Lake V1 boundary in blue (Arthur et al., 2025) and Vigrid_4 (this manuscript) upstream of Vigridisen.

- In the Discussion (Section 4.3), I suggest mentioning the lake cluster upstream of Jutulstraumen (Jutulstraumen_69, _71, _74), which you detect filling and draining between 2013–2015 and 2017–2020. This cluster is located 81 km upstream of the active lake cluster reported by Neckel et al. (2021) using Sentinel-1 DInSAR, validated with ICESat-2, and observed to fill and drain between 2017–2020 (see right-hand map). Given their proximity and temporal activity, these lakes may form part of a larger interconnected network and could be associated with cascading drainage.

- One of the interesting findings from this new inventory is the detection of active lakes near the grounding line in certain locations (e.g. King Baudouin East_3, less than 4 km from the GL). I think this deserves further emphasis (for example, an additional figure highlighting some of these new examples) due to its important implications for subglacial water routing and the impact on local ice dynamics at the transition between grounded and floating ice in these regions.
- As a general point, some of the discussion about future work suggestions could be more suited to Section 4.5 (Future Outlook), for example the last few sentences of Section 4.1 and 4.2.
- It is good to see details of the subglacial lakes, drainage/filling periods and associated area and volume changes provided in the Supplementary Data. I would encourage the authors to consider making their code available to reproduce the results presented in this manuscript, in line with *Nature Communication* guidelines and to promote reproducibility.

Specific comments:

The Abstract refers to 25 clusters, and it is nice that you show individual lakes in Supplementary Figure 2, but it is also difficult to see how these relate to each other. Might be nice to have an additional figure highlighting the clusters and the relationships between the lakes.

L17: I would specific 'lake clusters' here.

L26: I'm not sure that we can definitively say that ice thicknesses prevent surface meltwater from reaching the bed, given that direct observations of this are currently lacking here. I think you also meant for this sentence to join with the next one?

L29: I suggest rewording this to 'Our understanding of subglacial melt rate magnitudes' rather than what they are.

L30: Which inaccessible region are you referring to here – Antarctica as a whole?

L69: There is some inconsistent use of the terms 'boundaries' vs 'perimeters' when referring to lake outlines, so please choose one and amend accordingly.

L85: Can you specify the magnitude of lake signal detectable with processed CryoSat-2 data (how small)?

L105: I would repeat the observation period here to help the reader: 'We use our 10-year elevation change map (Oct 2010 – July 2020) to identify (...)'.
L107: I suggest 'with a major axis' is slightly confusing and not needed – consider changing the sentence to 'areas of spatially consistent surface height change ≥ 5 km that were distinct from measurement noise'.

L122: Was the ± 0.3 m yr⁻¹ elevation change threshold selected based on previous approaches, or based on judgement after trying different thresholds or following a different approach?

L162 (Figure 1): I suggest rewording caption to 'Existing subglacial lake inventory is also shown'. I also think it would help the reader by labelling key regions as mentioned in the text, e.g. Wilkes Land, Victoria Land, Adelie and Bakutis Coasts, as well as major ice streams. Subglacial water flux should really be included in the Figure legend.

L172: I think it would be useful to refer to Figure 2 earlier in this paragraph, for example here.

L178: Specify that the median lake drainage volume is previously-reported.

L182 (Figure 2): Should Panel e (Moscow University 44) include a dashed line to indicate when it starts to fill? Also caption should read (...) compared to elevation change from non-subglacial lake regions (...)'.
L189: Refer to Fig. 2l here.

L196: The wording here is slightly confusing, I suggest changing to: 'where we observe quiescent periods during filling or draining'.

L221 (Table 1): Unclear to me why some lake volume change estimates are starred, please explain in caption.

L229: The meaning of this sentence is unclear, given the previous sentence.

L235: Suggest specifying how much area change.

L273 (Figure 3). Suggest adding small insets to each panel to help situate lake locations in Antarctica.

L350: Suggest it could be clearer to refer to 'activity rate patterns' as modes of activity, here and on L352.

L359: It seems unclear to me how you arrived at this point, as I can't find any mention of this or any correlation analysis with these variables. I see that ice speed and ice thickness for each lake are reported as part the supplementary dataset, although mean drainage flux is not. Could you elaborate further?

L360: Very long sentence, suggest splitting.

L368: This is mentioned as a general point above, but it is unclear what the 25 clusters are that are referred to, and what you mean by 'the same' modelled subglacial hydrological pathways.

L383: I suggest rewording the term 'length-scale' to, for example: 'indicates decadal variability in connected subglacial system'.

L412: It would be relevant to cite Neckel et al. (2021) here.

L434: By bed properties, are you referring to sediment saturation or other parameters?

L444: cycles and activity to me refer to similar things, so I would reword this to 'subglacial lake filling and draining'.

L462: Suggest changing 'pooling' to 'ponding'. Similarly, suggest changing 'pools' to 'lakes' on L466.

L464: Suggest changing 'activity variability' to 'filling and draining activity' for clarity.

Technical/minor corrections:

L17: no need to hyphenate 'subglacial'.

L25: Add comma after Antarctica.

L50: The methods' reliance (add apostrophe).

L51: New sentence: 'However, this is limited by (...)'.
'

L53: New sentence: 'However, these measurements (...)'.
'

L68: No need to hyphenate 10-years.

L79: No need to capitalise ice sheet.

L119: New sentence: 'In the event that the map is produced over the full period of the cycle, (...)'.
'

L136: Add 'measurements' after 'subglacial lake area'.

L137: New sentence: 'Therefore, we assume (...)'.
'

L142: New sentence: 'However, we note that (...)'. This sentence would also benefit from a citation.

L150: For brevity, I suggest rewording to: 'We identify 85 new active Antarctic subglacial lakes (Fig. 1) ...'.
'

L172: 37, not thirty-seven. Also suggest bracketing (subglacial lakes Whillans_180 and Scott_12).

L177: New sentence: 'This lake drained approximately 1.3 km³ (...)'.
'

L194: Suggest slight rewording for clarity: 'for example Totten_52, which was associated with 8 m of uplift'.
'

L201: Specify 'subglacial lake filling and draining episodes' here.

L260: Missing full stop. New sentence following line: 'Therefore, elevation change (...)'.
'

L263: Hyphenate 'decade long'.

L301: 'Adjacent to, but distinct from, the main Institute E1 lake area'. I think the following sentence is also unnecessary and could be deleted.

L307: Given the point made in the previous sentence, I suggest this sentence is unnecessary.

L314: Suggest this sentence can be more concise: 'Previous subglacial lake inventories document the existence of active subglacial lakes beneath fast ice streams alongside less active lakes towards the interior of the continent, often beneath slower flowing ice^{1, 11, 40, 41, 45, 47, 48}'.

L377: Suggest 'subglacial lake clusters in East Antarctica.' (...) Secondly, our results show (...)'.

L403: Change 'cascade' to cascading, also on L412 and 416.

L429: New sentence: 'Ice dynamics dominate mass loss in Antarctica, so it is important to quantify any impact from subglacial hydrology'.

L439: New sentence: 'Therefore, subglacial lake systems (...)'.

L451: New sentence: 'However, further observations are required so we can more accurately quantify the impact of drainage events on ice dynamic processes (...)'.

L454: Comma, not semicolon.

L474: New sentence: 'We provide the first time-evolving (...)'.

References:

Arthur, J.F., Shackleton, C., Moholdt, G., Matsuoka, K. and van Oostveen, J., 2025. Evidence of active subglacial lakes under a slowly moving coastal region of the Antarctic Ice Sheet. *The Cryosphere*, 19(1), pp.375-392, <https://doi.org/10.5194/tc-19-375-2025>.

Le Brocq, A.M., Ross, N., Griggs, J.A., Bingham, R.G., Corr, H.F., Ferraccioli, F., Jenkins, A., Jordan, T.A., Payne, A.J., Rippin, D.M. and Siegert, M.J., 2013. Evidence from ice shelves for channelized meltwater flow beneath the Antarctic Ice Sheet. *Nature Geoscience*, 6(11), pp.945-948, <https://doi.org/10.1038/ngeo1977>.

Neckel, N., Franke, S., Helm, V., Drews, R., and Jansen, D., 2021. Evidence of Cascading Subglacial Water Flow at Jutulstraumen Glacier (Antarctica) Derived From Sentinel-1 and ICESat-2 Measurements, *Geophys. Res. Lett.*, 48, e2021GL094472, <https://doi.org/10.1029/2021GL094472>.